# Pseudoirreversible inhibition elicits persistent efficacy of a sphingosine 1-phosphate receptor 1 antagonist

Yuya Maruyama[1,2,3], Yusuke Ohsawa[3], Takayuki Suzuki[3], Yuko Yamauchi[3], Kohsuke Ohno[3], Hitoshi Inoue[3], Akitoshi Yamamoto[3], Morimichi Hayashi[3], Yuji Okuhara[3], Wataru Muramatsu[1,2], Kano Namiki[1,2], Naho Hagiwara[1], Maki Miyauchi[1,2], Takahisa Miyao[1], Tatsuya Ishikawa[1,2], Kenta Horie[1], Mio Hayama[1,2], Nobuko Akiyama[1,2], Takatsugu Hirokawa[4,5] & Taishin Akiyama[1,2] ✉

Sphingosine 1-phosphate receptor 1 (S1PR1), a G protein-coupled receptor, is required for lymphocyte trafficking, and is a promising therapeutic target in inflammatory diseases. Here, we synthesize a competitive S1PR1 antagonist, KSI-6666, that effectively suppresses pathogenic inflammation. Metadynamics simulations suggest that the interaction of KSI-6666 with a methionine residue Met124 in the ligand-binding pocket of S1PR1 may inhibit the dissociation of KSI-6666 from S1PR1. Consistently, in vitro functional and mutational analyses reveal that KSI-6666 causes pseudoirreversible inhibition of S1PR1, dependent on the Met124 of the protein and substituents on the distal benzene ring of KSI-6666. Moreover, in vivo study suggests that this pseudoirreversible inhibition is responsible for the persistent activity of KSI-6666.

Sphingosine 1-phosphate (S1P), which is a metabolite of membrane sphingolipids, regulates cell migration, inflammatory response, angiogenesis, and neurogenesis[1]. The physiological functions of S1P are mediated by its binding to five subtypes of the sphingosine 1-phosphate receptor, which belongs to the G protein-coupled receptor (GPCR) family[1]. Among S1P receptors (S1PRs), S1P receptor 1 (S1PR1) is critical for lymphocyte trafficking and progression of immune and inflammatory responses. The binding of S1P to S1PR1 causes activation of Gi, a subunit of heterotrimeric guanine nucleotide-binding proteins (G proteins), leading to the inhibition of adenylate cyclase-induced cAMP synthesis and subsequent activation of several intracellular signaling pathways. The binding also induces the phosphorylation-dependent recruitment of β-arrestin and subsequent cellular internalization of the ligand–receptor complex.

Given its roles in lymphocyte trafficking, S1PR1 is a target for the treatment of autoimmune and inflammatory diseases. FTY720, also named fingolimod, is a first-generation S1PR1 modulator that was approved as a drug for the treatment of multiple sclerosis (MS)[2,3]. Mechanistically, FTY720 is phosphorylated in vivo into FTY720-phosphate (FTY720-P), which serves as a functional antagonist of S1PRs including S1PR1[2]. Thus, the binding of FTY720-P activates S1PR1 signaling, leading to the internalization and subsequent degradation of S1PR1. As a result, administration of FTY720 causes long-lasting inhibition of S1PR1 signaling through the depletion of cell surface S1PR1. However, in addition to lymphocyte trafficking, S1PR1 controls the function of cardiac cells[4]. Therefore, the agonistic activity of FTY720-P provokes bradycardia[4–6], a temporal reduction of the heart rate, as an undesirable side effect.

[1]Laboratory for Immune Homeostasis, RIKEN Center for Integrative Medical Sciences, Yokohama 230-0045, Japan. [2]Immunobiology, Graduate School of Medical Life Science, Yokohama City University, Yokohama 230-0045, Japan. [3]Central Research Laboratory, Kissei Pharmaceutical Co., Ltd., 4365-1 Hotaka-Kashiwabara, Azumino, Nagano 399-8304, Japan. [4]Division of Biomedical Science, Faculty of Medicine, University of Tsukuba, 1-1-1 Tennodai, Tsukuba, Ibaraki 305-8575, Japan. [5]Transborder Medical Research Center, University of Tsukuba, 1-1-1 Tennodai, Tsukuba, Ibaraki 305-8575, Japan. ✉e-mail: taishin.akiyama@riken.jp

Competitive antagonists of S1PR1 would not cause side effects ascribed to the agonistic activity. W146, which has a similar structure to FTY720-P, was the first competitive antagonist of S1PR1[7,8]. However, the in vivo efficacy of W146 is poor and transient[8]. In addition to FTY720-based S1PR1 modulators such as W146, structurally distinct S1PR1 modulators have been developed[9]. Among them, NIBR-0213 is a competitive antagonist of S1PR1 that displayed persistent efficacy in vivo[10]. Further development of potent S1PR1 antagonists may offer more effective methods of treatment for autoimmune diseases such as MS.

The efficacy of a drug is affected by its residence time in the target protein in addition to its equilibrium-binding affinity and pharmacokinetics[11–13]. Therefore, the determination of drug–protein binding kinetics is useful for predicting drug efficacy[14]. Some studies have revealed that certain antagonists exhibit insurmountable antagonism[15–17], defined as the inability of the pre-bound antagonist to be completely replaced by a competing ligand under nonequilibrium conditions[18]. Such inhibitors are considered to possess pseudoirreversible inhibitory activity, characterized by a slow dissociation from the receptor without the formation of a permanent covalent bond for inhibition[19]. Indeed, it was reported that drugs with pseudoirreversible inhibition properties show persistent in vivo efficacies because of their long residence time[20].

Computational approaches can be used to investigate the binding behavior of drugs to target proteins[21–27]. Molecular dynamics simulations can be used to evaluate binding and release pathways, kinetics, and thermodynamics[28]. Metadynamics (MetaD), which is a sampling method enhanced by escaping local free-energy minima[25,29], and other approaches[25] are available for a detailed description of binding and unbinding kinetics[21,23,24,26]. However, the linking of these computational estimations for binding and dissociation kinetics to practical in vitro and in vivo efficacies of a drug remains a challenge.

Here, we show that the potent S1PR1 antagonist KSI-6666 achieves pseudoirreversible inhibition of S1PR1 activity by interacting with a specific methionine residue. KSI-6666 demonstrates potent and sustained efficacy in in vivo mouse disease models. In vitro studies reveal that KSI-6666 functions as a pseudoirreversible inhibitor. A combination of MetaD simulation and experimental validation suggests that this pseudoirreversible inhibition depends on the interaction between a methionine residue within S1PR1 and KSI-6666, accounting for the persistent potency of KSI-6666 in vivo.

## Results

### KSI-6666 is a potent and selective competitive antagonist of S1PR1 and suppresses autoimmune and intestinal inflammation

To develop new competitive antagonists of S1PR1, we synthesized 1181 compounds with structures similar to previous S1PR1 inhibitors. Subsequently, we screened these compounds by evaluating their antagonistic effects through in vitro GTP analog binding to Gi in cell membrane fractions obtained from S1PR1-expressing human embryonic kidney (HEK293) cells treated with an S1PR1 agonist. Through this process, we identified the lead compound 1 (Fig. 1a), bearing similarities to a compound previously discovered by Novartis. Further optimization of lead compound 1 was conducted using the GTP binding assay screening, employing the S1PR1 agonist from Merck (Merck S1PR1 agonist)[30] (Fig. 1a). In addition, the reduction in blood leukocyte count induced by inhibition of S1PR1 after the oral administration of compounds was evaluated in in vivo screening in rats (Fig. 1b). The selection identified a S1PR1 antagonist (RS)-KSI-6666 (hereafter referred to as KSI-6666 for simplicity), a racemic form with a half-maximal inhibitory concentration ($IC_{50}$) of 6.4 nM in GTP binding assay. Reduction in blood leukocyte count caused by KSI-6666 was observed at 48 h after administration (Fig. 1b), and thus it may be more persistent than NIBR-0213 since its response duration was reported as

24 h at 30 mg/kg administration[10]. Consequently, we concluded that KSI-6666 is a potent antagonist with persistent efficacy.

We further confirmed that KSI-6666 had no agonistic activity on S1PR1 signaling. The binding of an agonist to S1PR1 results in the activation of Gi and receptor internalization[1]. Consistently, FTY720-P and the Merck S1PR1 agonist led to an increment in the GTP binding to Gi and a decrease of surface S1PR1 on HEK293 cells (Fig. 1c, d). In contrast, treatment with KSI-6666 did not activate S1PR1 (Fig. 1c, d), confirming its lack of agonistic activity. Interestingly, the KSI-6666 treatment led to a decrease in GTP binding activity and an increase in the surface amount of S1PR1, even in the absence of agonists (Fig. 1c, d). This suggests its ability to suppress the constitutive activation of S1PR1. Thus, KSI-6666 exhibits the characteristics of an inverse agonist, defined as a compound that elicits pharmacological activity contrary to that of the agonist by binding to the same receptor[31], in addition to the competitive inhibitory activity. In the case of W146, inverse agonistic activity was detected at only high concentrations (Fig. 1d).

We next tested the antagonistic activity of KSI-6666 toward ligand binding to S1PR1 in other assays. S1PR1 activation increases $Ca^{2+}$ mobilization in cells over-expressing Gα15, a subunit of G protein. KSI-6666 competitively inhibited the agonist-dependent $Ca^{2+}$ mobilization of S1PR1, but not for other S1PRs (Fig. 1e and Supplementary Fig. 1). KSI-6666 inhibited mouse and rat S1PR1 with similar inhibitory activity as human S1PR1, demonstrating its cross-species efficacy (Fig. 1e and Supplementary Fig. 1). Furthermore, the agonist-dependent internalization of S1PR1 on HEK293 cells was antagonized by KSI-6666 in a dose-dependent manner (Fig. 1f). Pretreatment with KSI-6666 completely blocked the reduction of heart rate induced by administration of FTY720 in the in vivo experiment using guinea pigs (Fig. 1g), which most reflects the human situation[32]. These data suggest that KSI-6666 is a selective and competitive antagonist of S1PR1 in mammals, and, therefore, would not cause bradycardia as a side effect.

S1PR1 signaling is critical for lymphocyte trafficking, and therefore, the inhibition of S1PR1 impairs lymphocyte migration from lymphoid organs to the bloodstream, thereby causing a decrease in blood lymphocyte number. In Transwell assays in vitro, the addition of KSI-6666 inhibited the motility of murine splenocytes induced by S1P (Fig. 1h). Consistently, the administration of KSI-6666 effectively reduced blood lymphocyte numbers in rats (Fig. 1i, upper) and cynomolgus monkeys (Fig. 1i, lower), confirming its efficacy in both primates and rodents. Moreover, flow cytometric analysis showed that the numbers of CD4-, CD8-, and CD19-positive cells in mouse blood were significantly reduced 4 h after intravenous injection of KSI-6666 (Fig. 1j). In contrast, injection of W146 caused a slight reduction of B cell number in the blood at 1 h, but not at 4 h (Fig. 1j). The data suggest that, compared with W146, KSI-6666 has a potent and persistent effect in suppressing lymphocyte trafficking in rodents and primates.

In addition to the lymphocyte trafficking, S1PR1 modulators have a common effect on the endothelial barrier[7]. Consistently, we observed that the repetitive oral administration of KSI-6666 caused vascular hyperpermeability-related effects such as edema in the lungs (Supplementary Fig. 2a, b), further supporting that KSI-6666 can inhibit the functions of S1PR1 in vivo. Notably, although edema was observed following the administration of both KSI-6666 and FTY720, lung function parameters, respiratory rate (f) and respiratory pressure curves (Penh), were minimally affected by KSI-6666 in comparison to FTY720 (Supplementary Fig. 2c), suggesting a lesser impact of KSI-6666 on lung function.

These data strongly suggested that KSI-6666 functions as a potent competitive inhibitor of S1PR1 with less effect on cardiac and pulmonary functions. We then investigated the therapeutic potential of KSI-6666 in autoimmune and inflammatory diseases by using rodent models. In the rat experimental autoimmune encephalomyelitis (EAE) model,

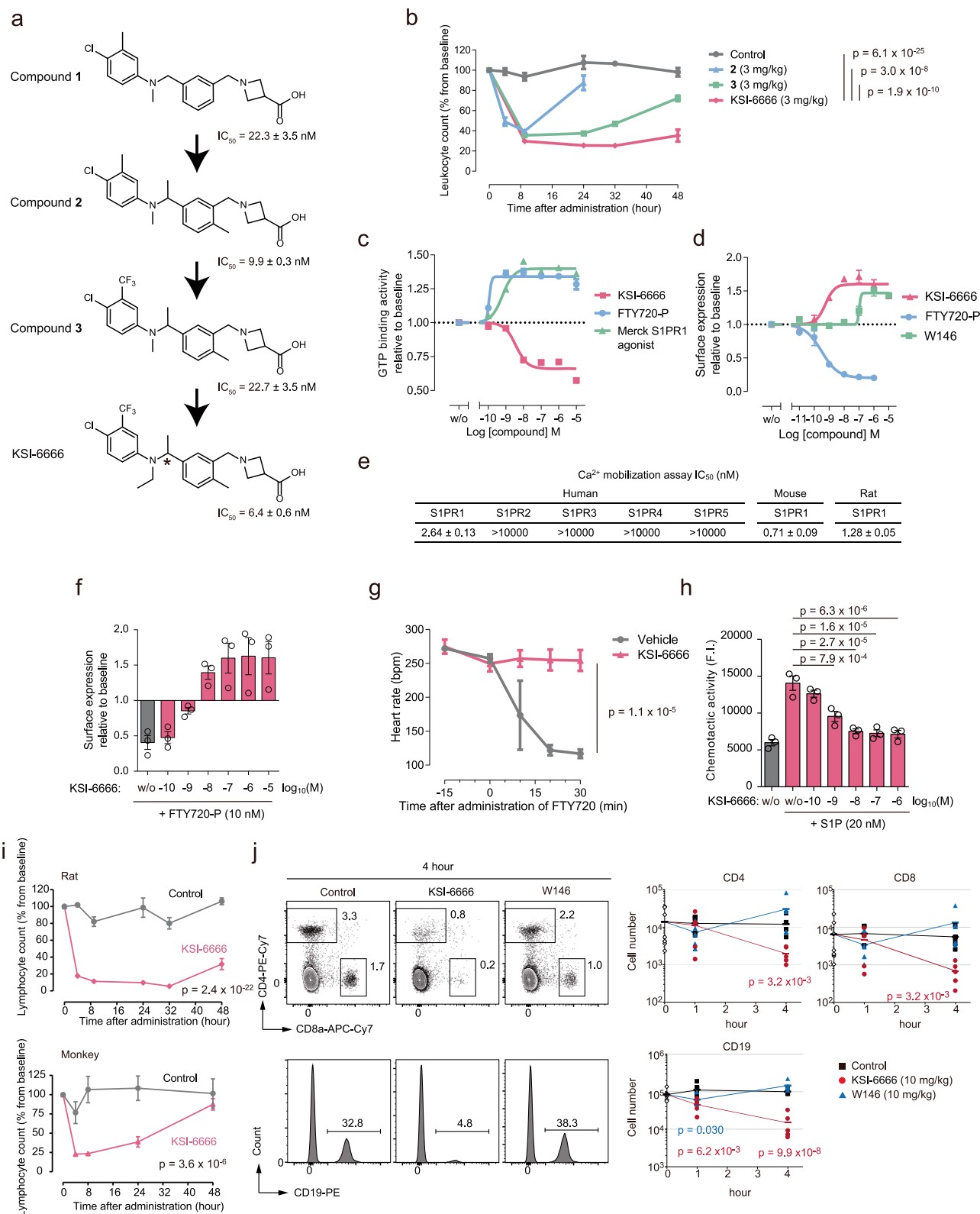

pre-administration of KSI-6666 demonstrated a significant attenuation of symptom progression (Fig. 2a). Furthermore, post-administration of KSI-6666 after disease onset resulted in a reduction in symptom scores in the mouse EAE model (Fig. 2b). Notably, the efficacy of KSI-6666 in ameliorating symptoms was comparable to that of FTY720.

In addition to the EAE model, studies in a T-cell transfer colitis model (Fig. 2c and Supplementary Fig. 2d) revealed that administration of KSI-6666 significantly suppressed the induced intestinal inflammation,

comparable with the effect of KRP-203, a functional agonistic inhibitor of S1PR1[33]. Consistently, the administration of KSI-6666 suppressed the expression of the chronic inflammatory marker S100a9 in the intestine[34]. Moreover, in an oxazolone-induced colitis model, KSI-6666 inhibited intestinal inflammation and up-regulated IL-4 protein levels in the whole intestine (Supplementary Fig. 2e). These findings collectively underscore the therapeutic potential of KSI-6666 in alleviating symptoms associated with autoimmune and inflammatory conditions.

**Fig. 1 | KSI-6666 persistently antagonizes S1PR1. a** Lead optimization of S1PR1 antagonists. The $IC_{50}$ values in the GTP binding assay are shown ($n = 3$ biological replicates; mean ± s.e.m.). An asterisk indicates a chiral carbon atom. **b** Effect of orally administered KSI-6666, compounds 2 and 3 on leukocyte number in the blood of Sprague-Dawley (SD) rats. Leukocyte number was normalized as a percentage from the baseline count before administration (control, $n = 6$ rats; compound 2, $n = 4$ rats; compound 3, $n = 4$ rats; KSI-6666, $n = 4$ rats; mean ± s.e.m.; two-way analysis of variance (ANOVA) vs. KSI-6666). **c** Effect of KSI-6666, FTY720-phosphate (FTY720-P), and Merck S1PR1 agonist on GTP binding activity ($n = 3$ biological replicates; mean ± s.e.m.). **d** Effect of KSI-6666, FTY720-P, and W146 on surface S1PR1 on HEK293 cells ($n = 3$ biological replicates; mean ± s.e.m.). **e** The $IC_{50}$ values of KSI-6666 across S1PR subtypes in $Ca^{2+}$ mobilization assay ($n = 3$ biological replicates; mean ± s.e.m.). **f** Effect of KSI-6666 pretreatment on downregulation of surface S1PR1 on HEK293 cells induced by FTY720-P. w/o indicates no treatment

with KSI-6666 ($n = 3$ biological replicates; mean ± s.e.m.). **g** Effect of KSI-6666 pretreatment on the FTY720-P-induced reduction in the heart rate of guinea pigs ($n = 3$ animals /group; mean ± s.e.m.; two-way ANOVA). **h** Effect of KSI-6666 pretreatment on the FTY720-P-induced chemotactic activity of splenocytes in transwell assay ($n = 3$ animals; mean ± s.e.m.; one-way ANOVA followed by Dunnett's test vs. treatment with S1P alone). **i** Effect of orally administered KSI-6666 (3 mg/kg) on lymphocyte number in the blood of rats (upper) and cynomolgus monkeys (lower). Lymphocyte number was normalized as a percentage from the baseline count before administration (control, $n = 3$ animals; KSI-6666, $n = 4$ animals in rat study; $n = 3$ monkeys /group in monkey study; mean ± s.e.m.; two-way ANOVA vs. vehicle). **j** Flow cytometric analysis of murine blood $CD4^+$, $CD8^+$, and $CD19^+$ lymphocytes after intravenous administration of KSI-6666 and W146 (10 mg/kg). $n = 6$ mice each for 1 and 4 h, $n = 9$ mice for 0 h (mean; two-tailed Student's $t$-test vs. vehicle). Source data are provided as a Source Data file.

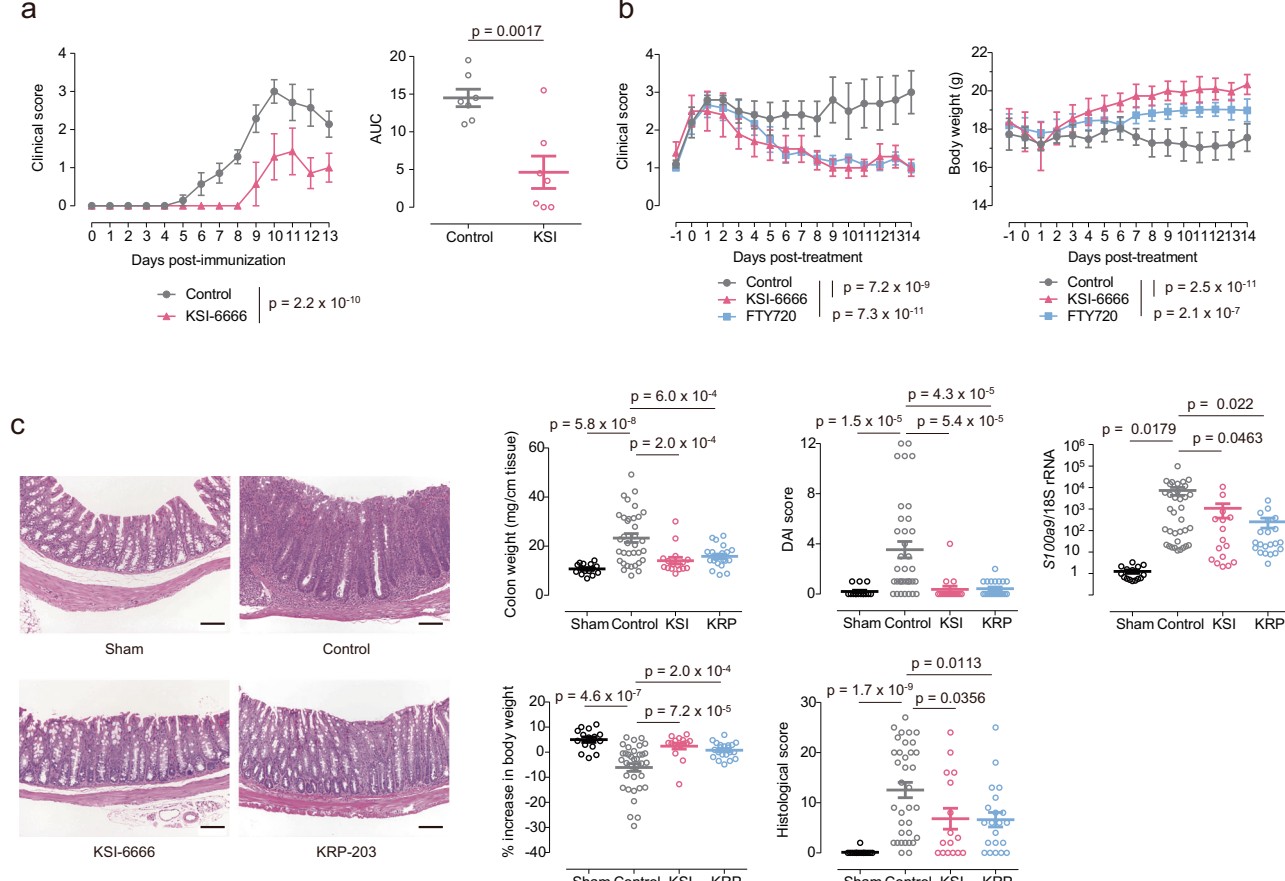

**Fig. 2 | KSI-6666 is a potent anti-inflammatory agent. a** Prophylactical efficacy of KSI-6666 (15 mg/kg/day) in rat EAE model. Rats were immunized by the intradermal injection of the syngeneic CNS antigen emulsified in complete Freund's adjuvant. On the next day of the immunization, oral administration of compounds twice daily was started and lasted for 12 days. The clinical score and area under curve (AUC) of the score over time were evaluated in each group. Statistical analyses vs. vehicle of clinical score and AUC were performed by two-way ANOVA and two-tailed Student's $t$-test, respectively ($n = 7$ rats/group; mean ± s.e.m.). **b** Therapeutic efficacies of KSI-6666 (30 mg/kg/day) and FTY720 (3 mg/kg/day) in mouse EAE model. C57BL/6JJcl mice were immunized by the subcutaneous injection of myelin oligodendrocyte glycoprotein peptide ($MOG_{35-55}$), followed by the intraperitoneal injection of the pertussis toxin. The day after the first observation of clinical score 1 or above in each animal, oral administration of compounds once daily was started and lasted for 14 days. The clinical score and body weight were evaluated in each group. Statistical analyses vs. vehicle were performed by two-way ANOVA (control,

$n = 5$ mice; KSI-6666, $n = 5$ mice; FTY720, $n = 6$ mice; mean ± s.e.m.). **c** Therapeutic efficacies of KSI-6666 (15 mg/kg/day) and KRP-203 (1.5 mg/kg/day) in the T-cell transfer colitis model. Isolated naive $CD4^+CD45RB^{high}$ T cells from BALB/cA mice were injected intraperitoneally into SCID mice and colitis was elicited. Two weeks after the transfer, twice daily oral administration of each compound was started and lasted for 14–16 days. Animals were sacrificed on the day after the final administration. Representative images of the histology were exhibited in left panels (scale bar = 100 μm). Colon weight, disease activity index (DAI) score, histological score, S100a9 mRNA expression level relative to ribosomal RNA expression, and % increase in body weight (between the day compound administration started and the end of the study) were evaluated in each group. Statistical analysis was performed by either two-tailed Student's $t$-test or two-tailed Welch's $t$-test according to $F$-test results (sham, $n = 15$ mice; control, $n = 35$ mice; KSI-6666, $n = 16$ mice; KRP-203, $n = 21$ mice; mean ± s.e.m.). Source data are provided as a Source Data file.

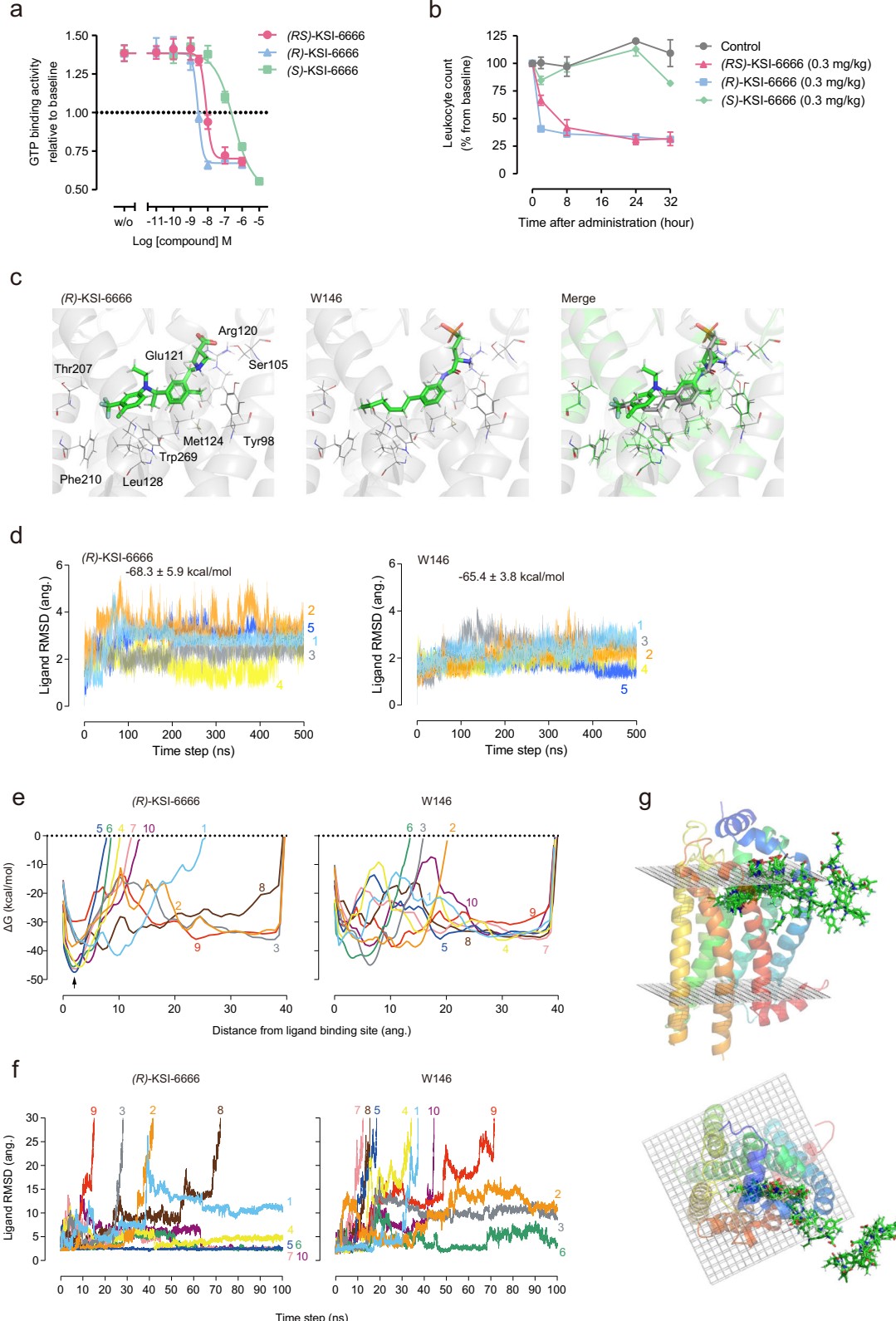

## KSI-6666 dissociates from S1PR1 with slower kinetics compared to W146

The structure of KSI-6666 is largely divisible into a zwitterionic part consisting of a carboxylic acid and an aliphatic tertiary amine, and a hydrophobic moiety consisting of two aromatic rings with bulky substituents. Additionally, there is a chiral carbon atom in the linker, situated between the two aromatic rings (marked with an asterisk in Fig. 1a).

Among the enantiomers, the S1PR1 inhibitory activity of the (R)- KSI-6666 was more potent than that of the (S)- KSI-6666 (Fig. 3a, b). This suggests that the spatial arrangement of the distal benzene ring may be important for the potent binding and inhibitory activity toward S1PR1.

To gain more insight into the structural basis of the activity of KSI-6666, we first performed molecular docking analysis for the binding of (R)-KSI-6666 with S1PR1. Co-crystallographic data for the complex of

**Fig. 3 | Metadynamics simulation suggests the dissociation process of KSI-6666 from S1PR1. a** Difference between KSI-6666 enantiomers in inhibitory activities against GTP binding to S1PR1 induced by 10 nM Merck S1PR1 agonist. Data were obtained from three independent experiments. Results are expressed as the mean ± s.e.m. Magenta line, (RS)-KSI-6666; cyan line, (R)-KSI-6666; green line, (S)-KSI-6666. **b** Effect of (RS)-KSI-6666, (R)-KSI-6666, and (S)-KSI-6666 on leukocyte number in blood of rats. Each compound was orally administered to SD rats at 3 mg/kg. Leukocytes in the blood were counted at the indicated times. Leukocyte number was normalized as a percentage from the baseline count before the administration (*n* = 3 rats /group; mean ± s.e.m.). **c** Docking pose of (R)-KSI-6666 and crystallographic pose of W146 with S1PR1. In the merged figure, (R)-KSI-6666 and W146 are depicted in green and gray, respectively. **d** Ligand RMSD (root mean square deviation) values during conventional molecular dynamics (cMD) simulations of (R)-KSI-6666- and W146-bound S1PR1. The plot shows the temporal changes in the structural stability of the ligand binding over the course of the molecular dynamics simulation. **e** Plots of binding free-energy profiles (kcal/mol) against the distance between the center of mass of the ligand-binding residues and the ligand in metadynamics (MetaD) trajectories for (R)-KSI-6666 and W146 in the dissociation pathway from S1PR1. The results from all ten simulation runs are plotted for each compound. **f** Plots of ligand root mean-square deviation against simulation time in the MetaD trajectories for (R)-KSI-6666 and W146 in the dissociation pathways from S1PR1. The results from all ten runs are plotted for each compound. **g** Representative trajectory images showing the dissociation path of (R)-KSI-6666 from S1PR1 predicted from metadynamics simulation. Sequential snapshots (every 10 ns of two representative runs) capturing key intermediate stages in the dissociation process are exhibited. (R)-KSI-6666 undergoes dissociation from the spatial region situated between Helix I (highlighted in blue) and Helix VII (highlighted in red) within the structure of S1PR1. Source data are provided as a Source Data file.

W146 and S1PR1 (PDB: 3V2Y)[35] were used for the docking study of (R)-KSI-6666. The pose of (R)-KSI-6666 with the lowest binding free energy was selected from among the docking complexes, and compared with the crystallographic structure of W146 (Fig. 3c). The validity of the docking pose was assessed through 500 ns MD simulations, demonstrating its stability (Fig. 3d). The estimated binding free energy to S1PR1 was −68.3 kcal/mol for (R)-KSI-6666 and −65.4 kcal/mol for W146, supporting stable binding of (R)-KSI-6666. Furthermore, molecular docking analysis using the recently determined S1PR1 structure from cryogenic electron microscopy[36] as the initial structure supported the binding of KSI-6666 on the ligand binding site of S1PR1 (Supplementary Fig. 3). Overlay of the pose of (R)-KSI-6666 with the W146 structure suggested that the two compounds would adopt a similar pose in the ligand-binding pocket. Thus, a polar group (a carboxylic acid in (R)-KSI-6666 and a phosphate in W146) and a benzene ring (the internal benzene ring of (R)-KSI-6666 and the benzene ring of W146) overlapped, and the end of the hydrophobic moiety ($CF_3$− in (R)-KSI-6666 and $CH_3$− in W146) was in a similar position for the two inhibitors.

We next performed MetaD simulation, which is used for biasing toward dissociation by applying external forces, to explore and compare the dissociation process from S1PR1 between (R)-KSI-6666 and W146. In total, 10 simulation runs for 100 ns each were performed for each compound. A plot of binding free-energy profiles (kcal/mol) against the distance of (R)-KSI-6666 (or W146) from the binding site in enhanced sampling using MetaD simulation suggested that a stable binding state of (R)-KSI-6666, but not of W146, may exist around 2 Å away from the docking position (Fig. 3e). Moreover, to evaluate the dissociation process, the root mean-square distance (RMSD) of (R)-KSI-6666 and W146 from the docking position was plotted against the simulation time. Assuming that the ligand was dissociated from S1PR1 when the RMSD was >30 Å, dissociation of W146 occurred in 7 of 10 runs by 100 ns from the start of the simulation (Fig. 3f), but dissociation of (R)-KSI-6666 occurred in only 4 of 10 runs. The visible inspection of MetaD data suggested that these ligands dissociate between helices I and VII, which is the same with other ligands of S1PR1[35], supporting the validity of the MetaD simulation (Fig. 3g). Interestingly, in four runs for (R)-KSI-6666, the RMSD converged at around 2 Å after 100 ns (Fig. 3f), which is consistent with the binding free-energy profiles (Fig. 3f). Overall, MetaD simulation suggested that (R)-KSI-6666 may be more resistant than W146 to dissociation from S1PR1.

To verify the findings from MetaD simulation, binding and dissociation properties of KSI-6666 were determined experimentally by the functional reversibility assay[37] (Fig. 4a). Thus, inhibitory effect of KSI-6666 was evaluated by its resistance against ligand-inducing receptor internalization (Supplementary Fig. 4). After S1PR1-expressing cells were pretreated with KSI-6666 at various concentrations, serially diluted amounts of the agonistic FTY720-P were added (Fig. 4b). The replacement of KSI-6666 with agonist FTY720-P on S1PR1 was monitored via the level of cellular surface S1PR1 (Fig. 4b). As

described above, in the absence of the agonist, the addition of KSI-6666 caused a reduction of baseline activity due to the inverse agonistic activity (labeled "w/o" in Fig. 4b). As expected, the addition of FTY720-P or S1P ligand caused the replacement of KSI-6666 in a dose-dependent manner. Of note, however, the maximum agonist effect ($E_{max}$) did not reach 100% when the pretreatment was with a high concentration of KSI-6666 (Fig. 4b). Furthermore, $E_{max}$ decreased with increasing concentrations of pretreated KSI-6666 (Fig. 4b right), but independent of the pre-incubation time (Supplementary Fig. 4b). This phenomenon occurred for both agonists, FTY720-P and the natural ligand S1P. Moreover, the increment of GTP binding activity on agonist addition did not reach the maximum in the presence of a high concentration of KSI-6666 (Fig. 4c). This indicates that the replacement of KSI-6666 by the agonists was not complete in the assay. In contrast, when W146 was used in the pretreatment, the agonistic responses reached the maximum level, suggesting the complete replacement of W146 by these agonists.

Due to the observed reduction in $E_{max}$, indicating a slower dissociation of KSI-6666 compared to W146, we conducted experiments to measure the dissociation half-life of KSI-6666 from S1PR1. In this experiment, pretreated cells with KSI-6666 were washed, followed by incubation in a medium for various times without KSI-6666. Subsequently, the agonistic ligand was added to monitor the free S1PR1 ratio, assessed through $Ca^{2+}$ mobilization (Fig. 4d). Data analysis revealed a dissociation half-life of 9.41 h for KSI-6666 and 0.20 h for W146 (Fig. 4d), confirming the very slow dissociation of KSI-6666 from S1PR1.

A previous mathematical modeling study proposed that the in vivo persistent efficacy of a drug depends on the residence time in its receptor when the dissociation half-life exceeds the pharmacokinetic half-life[38]. To elucidate the mechanism underlying the in vivo persistent efficacy of KSI-6666, we evaluated its pharmacokinetics. Data analysis suggested that the half-life time in blood was approximately 6.65 hours, which is notably shorter than the dissociation half-life (Fig. 4e). As a result, our data suggests that the slow dissociation of KSI-6666 from S1PR1 is a key factor contributing to its in vivo persistent efficacy. In sum, these data suggest that KSI-6666 is an insurmountable antagonist with pseudoirreversible inhibitory activity, which is consistent with the hindered dissociation of KSI-6666 predicted by MetaD simulation.

## Residue Met124 of S1PR1 is critical for the pseudoirreversible inhibition by KSI-6666

We next sought to determine the interaction responsible for the pseudoirreversible inhibition of S1PR1 by KSI-6666. When the MetaD trajectories of (R)-KSI-6666 were visually inspected, (R)-KSI-6666 appeared to intimately interact with hydrophobic residue Met124 of S1PR1, whereas W146 did not (Fig. 5a and Supplementary Movies 1, 2). To confirm this observation, 100 snapshots from each MetaD trajectory were selected at equal intervals and analyzed by using the protein–ligand interaction fingerprint (PLIF) tool implemented in

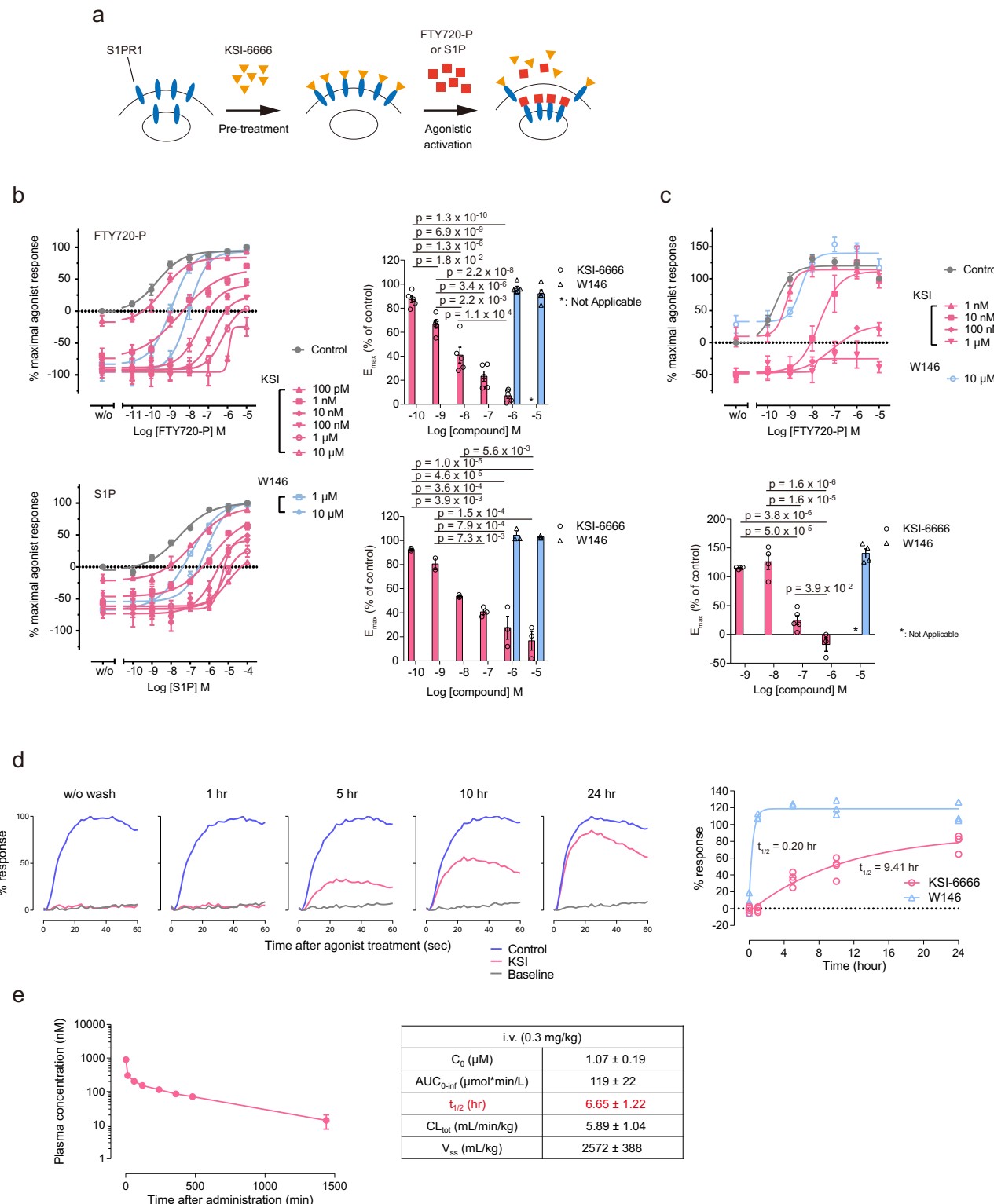

molecular operating environment (MOE) software, in which the interaction between a compound and each amino acid residue in a protein is represented as a fingerprint bit according to its interaction type. The frequency of bit entry for (R)-KSI-6666 relative to W146 was estimated. Data analysis suggested that, among the amino acids that (R)-KSI-6666 contacts during the dissociation process, Met124 might be the amino acid with the largest difference in contact frequency between (R)-KSI-6666 and W146, with (R)-KSI-6666 making more frequent surface contact with Met124 than doing W146 (Supplementary

Fig. 5a), implying a possible role of Met124 during the dissociation of (R)-KSI-6666 from S1PR1.

We experimentally verified the significance of Met124 for the interaction proposed from the MetaD analysis. HEK293 cells expressing mutant S1PR1 where Met124 was replaced by Val (Val124) or Leu (Leu124) were established (Supplementary Fig. 5b). Functional reversibility assays by measuring the internalization of cell surface S1PR1 showed that, in the Val124 mutant, pre-bound KSI-6666 was almost completely replaced by the added agonist (Fig. 5b), suggesting that the

**Fig. 4 | KSI-6666 is a pseudoirreversible inhibitor. a** Schematic figure of functional reversibility assay. **b** Functional reversibility of KSI-6666 and W146 examined by agonist-induced S1PR1 internalization assay. HEK293 cells expressing HiBiT-tagged human S1PR1 were treated with each compound and concentration-dependent S1PR1 internalization induced by FTY720-P or S1P, represented by luciferase activity, was measured. Calculated $E_{max}$ values for the FTY720-P and S1P-induced response on treatment with each compound at different concentrations are summarized in the right graphs. Data for the upper and lower graphs were obtained from five and three independent experiments, respectively. Statistical analysis was performed by one-way ANOVA, followed by Tukey's test for KSI-6666. Results are expressed as the mean ± s.e.m. **c** Functional reversibility of KSI-6666 and W146 examined by agonist-induced GTP binding assay. Calculated $E_{max}$ values for the FTY720-P-induced response on treatment with each compound at different concentrations. Data were obtained from eight independent experiments. Statistical analysis was performed by one-way ANOVA, followed by Tukey's test for KSI-

6666 (control, $n = 8$; 1 nM, $n = 4$; 10 nM, $n = 4$; 100 nM, $n = 5$; 1 μM, $n = 4$ biological replicates; mean ± s.e.m.). **d** Dissociation kinetics of KSI-6666 from S1PR1. KSI-6666 (30 nM) or W146 (10 μM) was incubated with cells expressing human S1PR1 for 60 min. After the removal of compounds through a washing process, cells were allowed to incubate for an indicated period to facilitate the dissociation of compounds from S1PR1. The cells were stimulated with 10 nM Merck S1PR1 agonist, and $Ca^{2+}$ mobilization was induced. The typical $Ca^{2+}$ mobilization responses are shown in the left panels. The relative response was plotted against the incubation time in the right figure, and the complex half-life time was evaluated. The results obtained from the operation without washout were plotted at 0 h. Data on KSI-6666 and W146 was obtained from four and three independent experiments, respectively. **e** Pharmacokinetics of KSI-6666 in rats. KSI-6666 was intravenously administered to rats, and blood samples were chronologically collected. Results are expressed as the mean ± s.e.m ($n = 3$ rats). Source data are provided as a Source Data file.

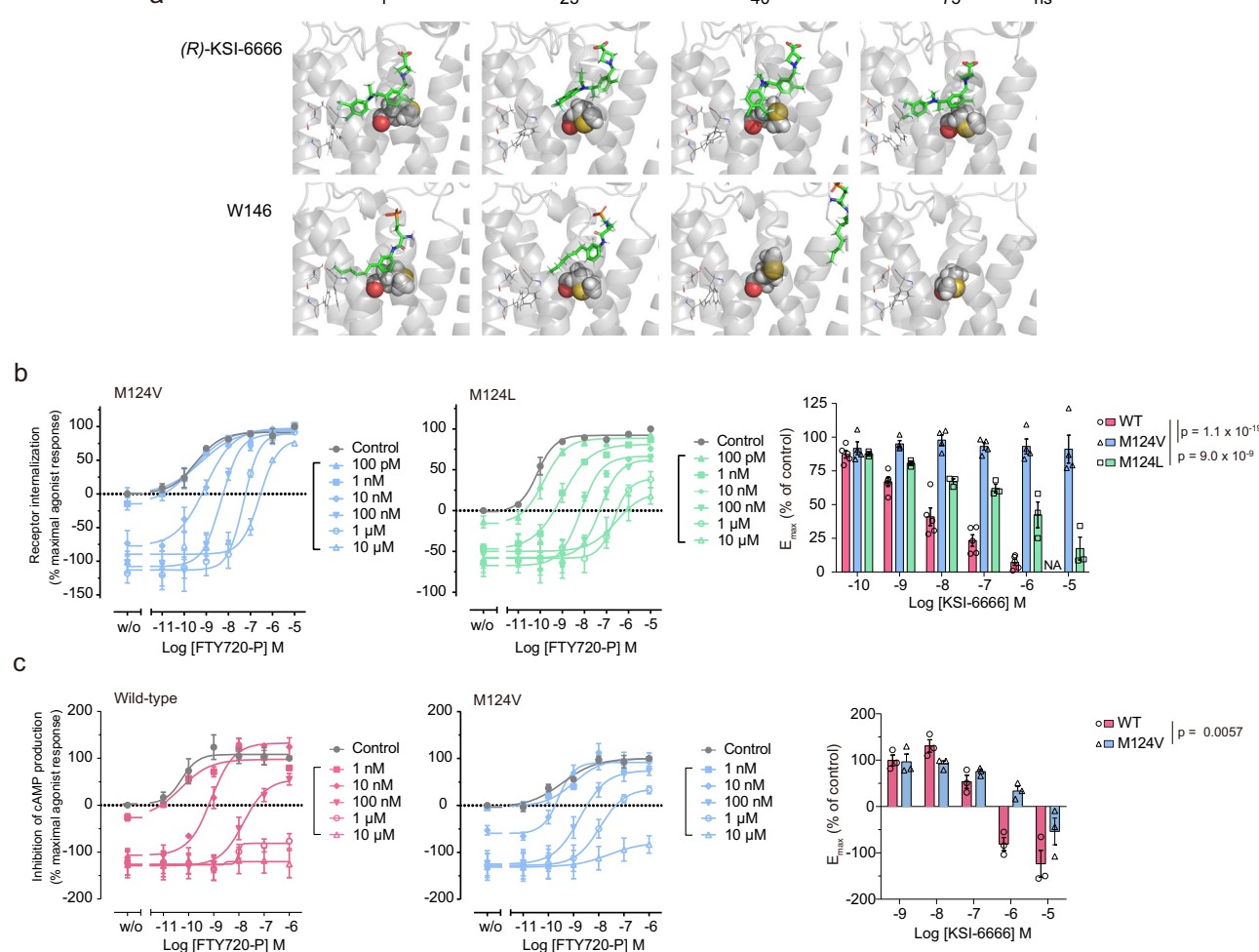

**Fig. 5 | Met124 of S1PR1 is critical for pseudoirreversible inhibition by KSI-6666. a** Typical poses of (R)-KSI-6666 in the ligand-binding pocket of S1PR1 in MetaD simulation. Residues Thr207, Phe210, and Phe273, representing the inner pocket, are shown in line style, and Met124 is shown in CPK style. **b** Functional reversibility of KSI-6666 with Val124 and Leu124 mutants of S1PR1 examined by the S1PR1 internalization assay. HEK293 cells expressing HiBiT-tagged human S1PR1 Val124 mutant (M124V) or Leu124 mutant (M125L) were treated with KSI-6666 and concentration-dependent receptor internalization stimulated by FTY720-P was measured. Calculated $E_{max}$ values for the FTY720-P-induced response on treatment with KSI-6666 at different concentrations are plotted for wild-type S1PR1 and the two mutants. The plot for the wild-type shown in Fig. 4b is reiterated here as a control. Data for the Val124 and Leu124 mutants (M124L) were obtained from four

and three independent experiments, respectively. Statistical analyses vs. wild-type were performed by two-way ANOVA for the Val124 and Leu124 mutants. Results are expressed as the mean ± s.e.m. **c** cAMP assay evaluating the functional reversibility of KSI-6666 in wild-type S1PR1 and the Val124 mutant. HEK293 cells expressing HiBiT-tagged wild-type S1PR1 or the Val124 mutant (M124V) were treated with KSI-6666 and concentration-dependent inhibition by FTY720-P of forskolin-induced cAMP production was measured. Calculated $E_{max}$ values for the FTY720-P-induced response on treatment with KSI-6666 at different concentrations are plotted for the wild-type and the mutant in the right graph. Data were obtained from three independent experiments, respectively. Statistical analysis was performed by two-way ANOVA. Results are expressed as the mean ± s.e.m. Source data are provided as a Source Data file.

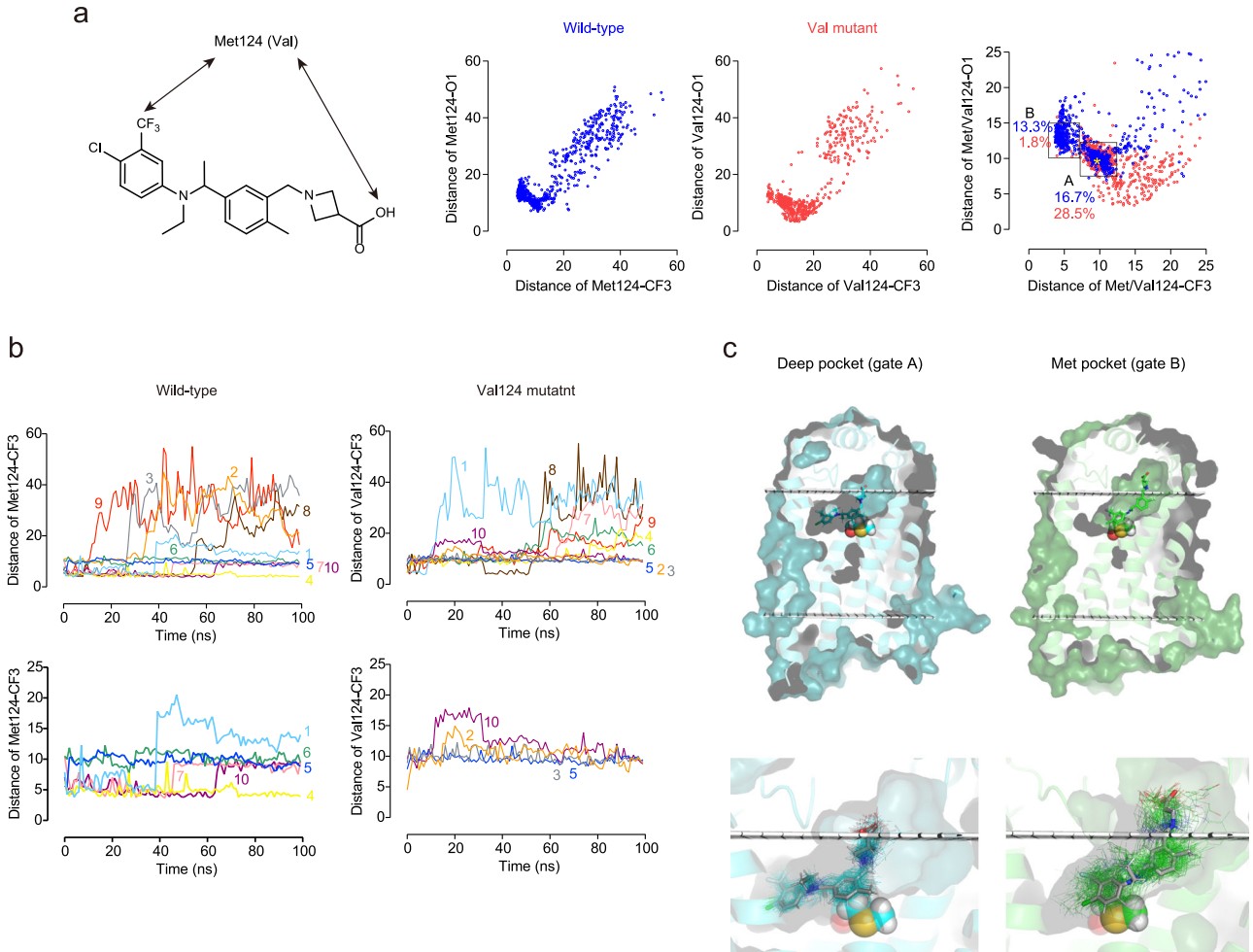

**Fig. 6 | Metadynamics simulation suggests the presence of a metastable binding pocket for KSI-6666 in S1PR1. a** Plot of the distance from Met124 (γ-carbon atom) to the carbon atom of the CF₃– group on the distal benzene ring of (R)-KSI-6666 against the distance from Met124 to the carboxylic acid (O1 atom) of (R)-KSI-6666 in MetaD snapshots, and corresponding plot of Val124 mutant. Snapshots ($n = 1000$) were selected every 1 ns from ten independent 100-ns MetaD simulations for each form of S1PR1. The area where both distances were <25 Å is enlarged in the right-hand figure. The frequency of poses assigned to gate A (distance from γ-carbon atom to the carbon atom of the CF₃– group: 2.5–7.8 Å; distance from γ-carbon atom to the O1 atom: 10–15 Å) and gate B (distance from γ-carbon atom to the carbon atom of the CF₃– group: 7.8–12.5 Å; distance from γ-carbon atom to the O1 atom: 7.5–12.5 Å) is shown for the wild-type (blue) and Val124 mutant (red) proteins. **b** Time-dependent changes in the distance between the carbon atom

of the CF₃– group of (R)-KSI-6666 and the γ-carbon atom of Met124 for wild-type or Val124 for M124V mutant of S1PR1. The results of all runs are shown in the upper figures. The results of runs in which the binding of (R)-KSI-6666 lasted for at least 100 ns are extracted in the lower figures. **c** The locations of the "deep pocket" and the "Met pocket" in S1PR1. The 100 snapshots per run structures of (R)-KSI-6666 (every 1 ns in 100-ns simulations) from Run 7 and 10 were picked up together and randomly clustered into two groups based on RMSD. These two groups corresponded to structures belonging to the deep pocket (gate A) and Met pocket (gate B). The stick model illustrates the average position of (R)-KSI-6666 in each pocket. The figures below offer an enlarged view of each pocket, and the line model indicates all structures of (R)-KSI-6666 positioned in each pocket. Met124 is shown in CPK style and the boundaries of the cell membrane are shown by white grids. Source data are provided as a Source Data file.

replacement of Met124 with Val abolished the pseudoirreversible inhibition by KSI-6666. The insurmountable inhibitory activity of KSI-6666 in agonist-induced cAMP production was also significantly impaired in the Val124 mutant, although the effect of the mutation was partial (Fig. 5c) as compared with assay monitoring of the S1PR1 amount (Fig. 5b). This may be attributable to the shorter incubation time in the cAMP production assay. Compared with the mutation to Val, the replacement of Met124 with Leu had a mild influence on the pseudoirreversible inhibition of KSI-6666 (Fig. 5b), highlighting the importance of the side chain property in Met124.

Met124 is critical for the pseudoirreversible inhibition, and therefore we next focused on the structural basis of the interaction of Met124 with (R)-KSI-6666 in MetaD simulation. Further visible inspection of the MetaD trajectories suggested that Met124 may interact with the distal benzene ring of (R)-KSI-6666, which has trifluorocarbon (CF₃–) and chloride (Cl–) substituents, rather than with the zwitterionic

end of the ligand (Fig. 5a and Supplementary Movie 1). To clarify this observation, the distance from Met124 (the γ-carbon atom) to the carbon atom of the CF₃– group was compared with that from Met124 to a hydroxyl oxygen atom of the carboxylic acid in the zwitterionic moiety (hereafter referred to as O1) for the all selected poses in the MetaD trajectories (Fig. 6a). In the original structure determined by the docking study (Fig. 3c), the distances from Met124 were 10.1 Å for CF₃– and 9.3 Å for O1. A correlation plot of the two distances in the simulation suggested that two major poses were present (named "gate A" and "gate B," respectively, see Fig. 6a) besides scattered unbound poses. Gate A has a median Met124–CF₃ distance of 9.42 Å and a median Met124–O1 distance of 9.97 Å, which is close to the values in the docking structure (yellow asterisk in gate A of Fig. 6a, right). Gate B has a median Met124–CF₃ distance of 6.27 Å and a median Met124–O1 distance of 12.95 Å. This suggests that, in the dissociation process of (R)-KSI-6666 during MetaD simulation, there may be another binding

structure in which the distal benzene ring in (R)-KSI-6666 possessing $CF_3-$ and $Cl-$ substituents becomes close to Met124.

To address whether the second binding structure depends on Met124, we performed the equivalent MetaD simulation for the Val124 mutant. Notably, the frequency of the poses in gate B decreased significantly in the MetaD simulation for the Val124 mutant (Fig. 6a). Overall, MetaD simulation suggested that a metastable binding state (hereafter referred to as the "Met pocket") may be generated by the interaction of (R)-KSI-6666 with Met124.

We next evaluated the time-dependent transition of (R)-KSI-6666 binding poses by plotting the distance between the $CF_3-$ group and Met124 in each run versus time. After starting the MetaD simulation from the docking position, (R)-KSI-6666 quickly moved to the Met pocket in seven runs (Runs 1, 3, 4, 7, 8, 9, and 10; Fig. 6b). Subsequently, (R)-KSI-6666 dissociated from the Met pocket in three runs (Run 3, 8, and 9); moved to another binding pocket close to the starting position (hereafter referred as to the "deep pocket") in two runs (Runs 7 and 10); and stayed in the Met pocket in one run (Run 4). In Run 1, although the distal benzene ring moved out of the pocket, (R)-KSI-6666 was still bound to S1PR1. In the other three runs (Runs 2, 5, and 6), (R)-KSI-6666 was located in the deep pocket soon after starting the simulation. Thereafter, (R)-KSI-6666 stayed in the deep pocket in two of the runs (Runs 5 and 6). Finally, in Run 2, (R)-KSI-6666 dissociated from the deep pocket. As expected, for the Val124 mutant, (R)-KSI-6666 stayed in the deep pocket after starting the simulation and dissociated without entering the Met pocket, except in Run 8, in which it may have been temporarily located in the Met pocket from around 30 to 50 ns (Supplementary Movie 3).

The deep pocket and Met pocket were visualized based on MetaD data (Fig. 6c). In the deep pocket, a large part of the (R)-KSI-6666 structure is located within the transmembrane region of S1PR1. In contrast, in the Met pocket, the zwitterionic part of (R)-KSI-6666 extends into the extracellular region of S1PR1. As expected, the distal benzene ring seems to closely associates with Met124 in the Met pocket, whereas in the deep pocket, the central benzene ring is in close proximity to Met124. Because the Met pocket appears to be positioned along the path of dissociating (R)-KSI-6666 from the deep pocket, its presence serves as a potential barrier to dissociation. Overall, these data suggest that Met124 may be involved in the formation of a metastable binding pocket, the Met pocket, for (R)-KSI-6666, and thereby could act as a "gatekeeper" to limit the dissociation of (R)-KSI-6666 from S1PR1.

### The pseudoirreversible inhibition contributes to the persistent efficacy of KSI-6666

Because the distal benzene ring of (R)-KSI-6666 may interact with Met124 in S1PR1 in the Met pocket, we experimentally tested if replacing substituents on the benzene ring with hydrogen atoms influenced the dissociation properties of KSI-6666. Notably, derivative compounds of KSI-6666 lacking the $CF_3-$ substituent (compounds 4 and 5 in Fig. 7a) were almost completely replaced with increased addition of FTY720-P in the functional reversibility assay monitoring the amount of surface S1PR1(Fig. 7b), suggesting no pseudoirreversible inhibitory activity for these compounds. Consistently, the dissociation half-life of compound 4 was determined to be 0.98 h (Fig. 7c), which was considerably faster than that of KSI-6666 (9.4 h in Fig. 4d). We next tested whether the in vivo efficacy of orally administered compounds 4 and 5 was altered compared with that of KSI-6666 by evaluation of their effects on blood leukocyte number. Notably, in contrast to KSI-6666, the reduction of blood leukocytes recovered to the basal level by 32 h after the administration of compound 4 or 5 (Fig. 7d left), even though the plasma concentrations of KSI-6666 and compound 4 were comparable (Fig. 7d right), indicating that replacement of the $CF_3-$ group with a hydrogen atom impaired the persistency of the efficacy of KSI-6666. Overall, the data suggest that pseudoirreversible inhibition elicits the persistent efficacy of KSI-6666 as an S1PR1 antagonist.

## Discussion

KSI-6666 showed persistent efficacy in vivo. Our data indicated that KSI-6666 is a pseudoirreversible inhibitor of S1PR1 due to its slow dissociation from the receptor. Previous studies proposed that slower dissociation of antagonists from receptors elicits longer and more potent inhibition of target molecules in vivo[38–41]. Notably, based on a model using six parameters of pharmacokinetics and dissociation kinetics, Dahl et al. proposed that a persistent efficacy of a drug due to long drug-target residence time can occur when the binding dissociation kinetic is slower than the pharmacokinetics elimination[38]. Data showed that the dissociation half-life of KSI-6666 from S1PR1 (9.41 h) was longer than that of the pharmacokinetics half-life (6.65 h). This suggests that the pseudoirreversible binding of KSI-6666 to S1PR1 would be a determinant in its persistent efficacy. Consistently, compound 4, which lacks substituents of the benzene ring in S1PR1, was not a pseudoirreversible inhibitor and did not exhibit persistent efficacy. However, it cannot yet be denied that the relatively long pharmacokinetics half-life of KSI-6666 also contributes to its persistent efficacy.

The two-step induced fit model[42] may explain the pseudoirreversible inhibition. In this model, stable binding of a drug with its target occurs via a two-step mechanism: in the first step, binding to the target forms an encounter complex; then, in the second step, a conformational change of the complex occurs and strengthens the drug binding to the target. In binding via the two-step induced fit mechanism, the residence time of a drug-target complex may be considerably augmented. MetaD simulation of KSI-6666 in the dissociation process from S1PR1 suggested the presence of another stable binding pose requiring residue Met124 (i.e., the "Met pocket") in addition to the pose close to the original binding pose (i.e., the "deep pocket"). The replacement of Met124 with Val abolished the pseudoirreversible inhibition by KSI-6666 in vitro. Therefore, KSI-6666 possibly binds to S1PR1 via a two-step induced fit mechanism; it binds to the Met pocket in the first step, and then forms a stable binding structure in the deep pocket, resulting in pseudoirreversible inhibition of S1PR1.

A previous study showed that the replacement of Met124 with Lys caused a severe reduction of S1PR1 activity in cell culture[43]. This finding suggests that the hydrophobicity of Met124 is critical for the formation of the binding pocket for the S1PR1 ligand. Our study showed that the replacement of Met124 with Val or Leu did not practically affect the activity of FTY720-P, but severely impaired the slow dissociation of KSI-6666. Thus, Met124 may play roles not only in the formation of a hydrophobic "wall" for the binding pocket but also as a "gatekeeper" for the binding and unbinding processes of S1PR1 modulators.

The benzene ring moiety in KSI-6666 appears to be critical for the interaction with Met124. S1PR1 modulators can be divided into two types: FTY720-based S1PR1 modulators, which have a flexible lipophilic chain, and structurally divergent ones, most of which have benzene rings[9]. The S1PR1 modulators with benzene rings may use the interaction with Met124 to form stable interactions, which could result in slow dissociation. Future testing of this hypothesis may help in the design of new S1PR1 antagonists.

S1PR modulator FTY720 has been used for the treatment of MS for more than a decade. FTY720 undergoes phosphorylated in the body, and phosphorylated form (FTY720-P) exerts its action through initial S1PR1 agonism and subsequent receptor downregulation, resulting in in vivo persistence of efficacy. The initial activation is known to cause bradycardia through the activation of S1PR1. Therefore, continuous monitoring of heart rate and blood pressure is recommended during the early phase after administration of FTY720. Another S1PR modulator, ozanimod, has fewer side effects and was approved for MS therapy[44]. However, ozanimod also has agonistic activity, thereby affecting heart function[45]. To avoid the side effects of S1PR modulator ascribed to the S1PR1 agonism, a competitive inhibitor of S1PR1 was

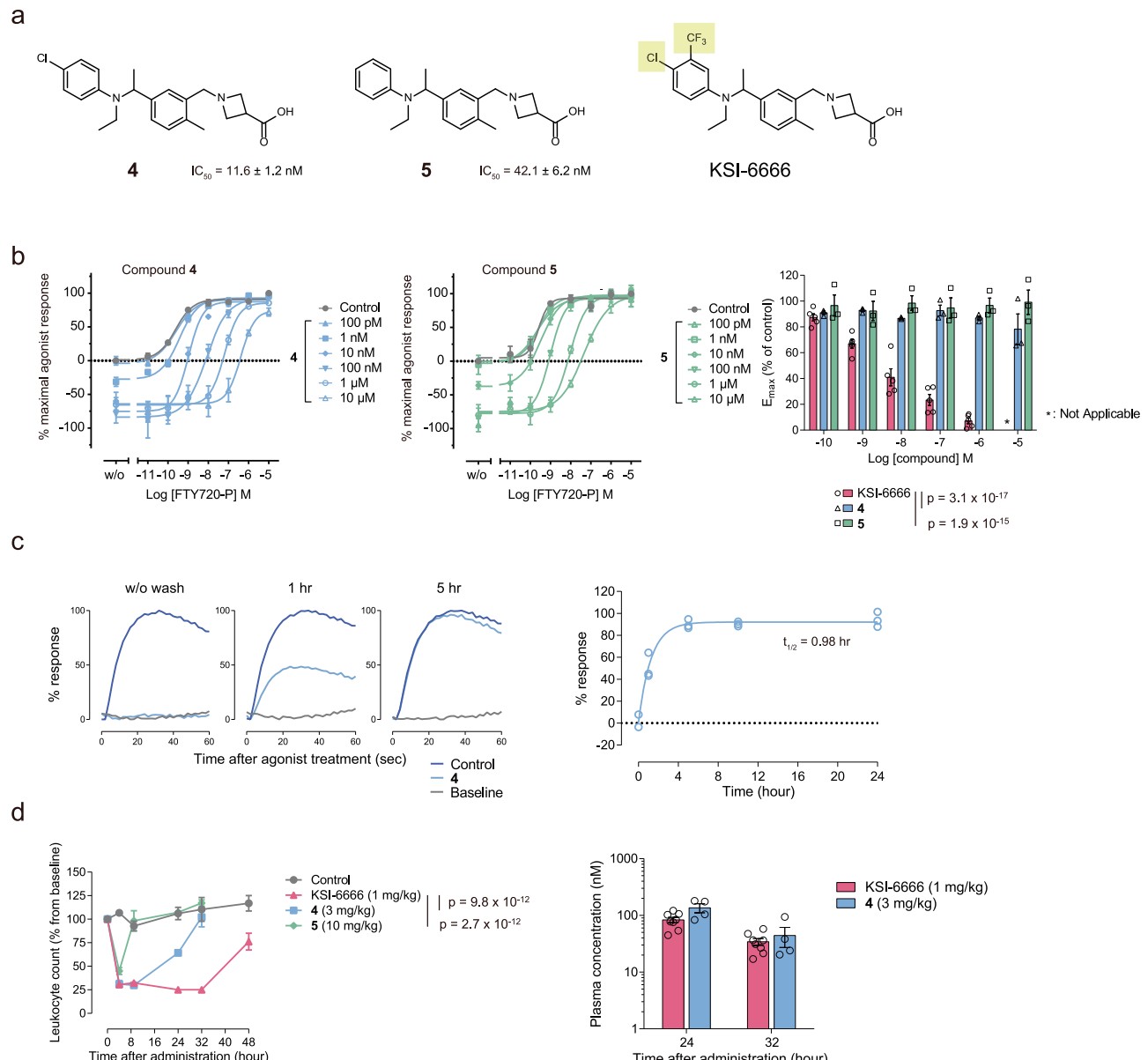

**Fig. 7 | Derivative compounds of KSI-6666 that lack pseudoirreversible inhibition have poor persistent efficacy. a** Structures of KSI-6666 and related compounds 4 and 5. **b** Functional reversibility of compounds 4 and 5 examined by S1PR1 internalization assay. HEK293 cells expressing HiBiT-tagged human S1PR1 were treated with compound 4 or 5 and concentration-dependent receptor internalization stimulated by FTY720-P was measured. Calculated $E_{max}$ values for the FTY720-P-induced response on treatment with these compounds at different concentrations are plotted. The plot of KSI-6666 data shown in Fig. 4b is reiterated as a control ($n = 3$ biological replicates; mean ± s.e.m.; two-way ANOVA vs. KSI-6666). **c** Dissociation kinetics of compound 4 from S1PR1. Compound 4 (30 nM) was incubated with cells expressing human S1PR1 for 60 min. After the removal of the compound through a washing process, cells were allowed to incubate for an indicated period to facilitate the dissociation of the compound from S1PR1. The cells were stimulated with 10 nM Merck S1PR1 agonist, and $Ca^{2+}$ mobilization was induced. The typical $Ca^{2+}$ mobilization responses are shown in the left panels. The

relative response was plotted against the incubation time in the right figure, and the complex half-life time was evaluated. The results obtained from the operation without washout were plotted at 0 h. Data were obtained from three independent experiments. **d** In vivo efficacy of orally administered KSI-6666, compounds 4 and 5 in SD rats. The efficacy, represented as the magnitude of reduction in blood leukocyte count, was evaluated. Leukocyte number was normalized as a percentage from the baseline count before administration (control, $n = 5$ rats; KSI-6666, $n = 8$ rats; compound 4, $n = 4$ rats; compound 5, $n = 4$ rats; mean ± s.e.m.; two-way ANOVA vs. KSI-6666). Data for a single individual belonging to the control shown in Fig. 1b are reiterated for comparison. Blood samples were not collected at 4 h from one and four animals in the control and KSI-6666 groups, respectively. Corresponding plasma concentrations of KSI-6666 and compound 4 at two time points are shown in the right-hand graph (mean ± s.e.m.). Source data are provided as a Source Data file.

developed. W146 was identified as a competitive antagonist of S1PR1[7] however, its persistency of efficacy was found to be low, as confirmed[8]. We here demonstrate that KSI-6666 is a competitive inhibitor with persistent efficacy in vivo, and its therapeutic efficacy on the mouse EAE model was comparable with FTY720. Notably, as expected, administration of KSI-6666 did not provoke bradycardia. Our study

suggested that the pseudoirreversible activity of KSI-6666 contributes to its persistent efficacy. In addition to KSI-6666, NIBR-0213 was previously identified as another competitive antagonist with a persistent reduction of blood lymphocyte count[10]. The authors concluded that the persistent effect of the NIBR-0213 is attributed to its long pharmacokinetics. Although the insurmountable activity of NIBR-0213 was

not reported in this study, NIBR-0213 may also be a pseudoirreversible inhibitor due to its chemical structure consisting of benzene ring moieties, similar to KSI-6666, which could interact with Met124 in S1PR1 during the dissociation process.

In addition to MS, new applications, such as treatment of ulcerative colitis, have been proposed for the S1PR1 agonists, ozanimod and etrasimod[46,47]. In the present study, KSI-6666, a competitive antagonist, showed an efficacy comparable to functional antagonists in rodent colitis models as well as the EAE model. Accordingly, competitive S1PR1 antagonists with pseudoirreversible inhibition, such as KSI-6666, could potentially be applicable for the treatment of these inflammatory diseases in future.

One limitation of this study is that the significance of Met124 was not demonstrated in vivo. Interestingly, an in vitro study suggested that the insurmountable activity of KSI-6666 was abolished in S1PR1, where Met124 was replaced by Val124. Since a genomic point mutation from A to G would change Met to Val, the in vivo effects of KSI-6666-like pseudoirreversible inhibitors, can be genetically variable. To confirm this, mice with a mutation in Met124 need to be established in future. Another limitation of our study is the absence of data regarding the dissociation kinetics and elimination kinetics in lymphoid organs, where KSI-6666 binds to S1PR1, inhibiting the egress of lymphocytes. Thus, in addition to the in vitro cell culture model, it is crucial to verify whether KSI-6666 exhibits slow dissociation from S1PR1 in a more physiological setting and to determine its elimination time from lymphoid organs. Consequently, conducting a detailed analysis of lymphoid organs from mice treated with KSI-6666 emerges as a significant avenue for future research.

Computational prediction of drug-binding behavior and free energy may be used widely in the development of new drugs in the future. In this study, we employed molecular dynamics to gain insights into the structural basis of the persistent efficacy of KSI-6666. The MetaD simulation highlighted the importance of Met124, a hypothesis substantiated through subsequent experiments involving cell cultures and animal models. A limitation of our study was the rudimentary nature of the MD simulation used to predict the dissociation behavior of KSI-6666. Although our simulation suggested the obstructive role of Met124 in the dissociation process of KSI-6666, it should be noted that a more detailed simulation is required to predict the dissociation kinetics quantitatively. Consequently, detailed and intensive computer simulations[21,23,48,49] would be more helpful for predicting the structural basis of drug binding and unbinding kinetics in addition to equilibrium-binding structures, which could aid in rational drug design in the future.

## Methods

### Animals
Animal experiments were performed in accordance with the protocols approved by the Laboratory Animal Committee of Kissei Pharmaceutical Co., Ltd. or the Guidelines of the Institutional Animal Care and Use Committee of RIKEN, Yokohama Branch (2018-075). BALB/cA mice (CLEA Japan and Jackson Laboratory Japan), BALB/cCr mice (Japan SLC), C57BL/6JJcl mice (CLEA Japan), C.B-17 SCID mice (Jackson Laboratory Japan), Sprague-Dawley (SD) rats (Japan SLC and Jackson Laboratory Japan), Dark Agouti (DA) rats (Japan SLC) and Hartley guinea pigs (Japan SLC) were maintained in standard controlled conditions with a 12-h lighting cycle and access to chow and water ad libitum. Cynomolgus monkeys (Del Mundo Trading and SICONBREC Inc.) were maintained in standard controlled conditions with a 12-h lighting cycle and access to chow from 9:00 to 13:00 and water ad libitum, and experiments were performed by Ina Research Inc. according to protocols approved by the Institutional Animal Care and Use Committee. In total, seven male monkeys were used for the study.

### Chemistry
KSI-6666, (R)-KSI-6666, (S)-KSI-6666, compounds 1, 2, 3, 4, and 5, the Merck S1PR1 agonist, W146, and KRP-203 were synthesized by Kissei Pharmaceutical Co., Ltd. Detailed synthetic schemes and synthetic procedures are presented in the Supplementary Methods in Supplementary Information file.

### Compound-induced reduction in blood leukocyte and lymphocyte count in rats
Compounds dissolved in the vehicle, which consisted of 5–10% dimethyl sulfoxide (Wako) and 5–10% Wellsolve (Celeste) mixed in distilled water (Otsuka), were orally administered to 6–16-week-old male SD rats. Blood samples were chronologically collected in EDTA-2K-containing blood collection tubes (BD) under anesthesia before and after compound administration. Leukocytes or lymphocytes in blood were counted by an automated hematology analyzer KX-21NV (Sysmex) or ProCyte Dx (IDEXX). In some experiments, plasma fractions were isolated and concentrations of compounds were measured by liquid chromatography-mass spectrometry (Q Exactive, Thermo Fisher Scientific). Detailed measurement procedures are presented in the Supplementary Methods in the Supplementary Information file.

### Compound-induced reduction in blood lymphocyte count in mice
Compounds were dissolved in 10 mM $Na_2CO_3$ and 40% (2-hydroxypropyl)-β-cyclodextrin (Sigma-Aldrich) and intraperitoneally administrated to 8-week-old female BALB/cA mice (10 mg/kg). At 1, 2, and 4 h after the administration, whole blood was collected into tubes pretreated with EDTA-2K (Dojin). After the lysis of erythrocytes by using RBC Lysis Buffer, cell suspensions were stained with anti-mouse CD19 (BD 557399), CD4-PECy7 (BioLegend 100528), and CD8-APCCy7 (BioLegend 100714) antibodies after blocking by anti-CD16/32 antibodies (BioLegend 101302). Dead cells were excluded via 7-aminoactinomycin D (Wako) staining. Flow cytometric analysis was performed by using Canto II (BD), and data were analyzed by Flowjo software (BD).

### Compound-induced reduction in blood lymphocyte count in cynomolgus monkeys
Three-to-five-year-old male cynomolgus monkeys were used. The test compound suspended in a vehicle consisting of 0.5% methylcellulose 400 solution (Wako) and 0.01 M hydrochloric acid (Wako) mixed in distilled water was orally administered to monkeys. Blood samples were chronologically collected in EDTA-2K-containing blood collection tubes while the monkey was awake, before and after compound administration. Lymphocytes in blood were counted using the hematology analyzer ADVIA120 (Siemens).

### Pharmacokinetic study in rats
Compounds dissolved in the vehicle, which consisted of 10% dimethyl sulfoxide and 10% Wellsolve mixed in saline (Otsuka), were intravenously administered to 6-week-old male SD rats. Blood samples were chronologically collected from awake rats after compound administration and heparinized. Plasma fractions were isolated and concentrations of compounds were measured by liquid chromatography-mass spectrometry (Q Exactive, Thermo Fisher Scientific). Detailed measurement procedures are presented in the Supplementary Methods in the Supplementary Information file. The pharmacokinetic parameters were calculated by non-compartmental analysis using WinNonlin 6.1 software (Pharsight).

### GTP binding assay
Membranes were prepared from HEK293 cells stably transfected with human S1PR1. GTP binding assay was performed using a GTP Gi

Binding Assay Kit (Cisbio) according to the manufacturer's protocol. In the antagonist assay, serial dilutions of the test compound and 10 nM Merck S1PR1 agonist were co-incubated with 1.25 μg/well of cell membranes suspended in the stimulation buffer supplied by the manufacturer containing 50 mM $MgCl_2$ and 2 μM GDP, followed by the addition of GTP Eu Cryptate (donor) and d2 antibody (acceptor). Membranes were incubated overnight at room temperature and the fluorescence signals from the donor (620 nm) and the acceptor (665 nm) were read using a PHERAstar FSX microplate reader (BMG Labtech). The fluorescence ratio (signal 665 nm/signal 629 nm × $10^4$) was calculated. The half-maximal inhibitory concentration ($IC_{50}$) values were calculated by nonlinear regression analysis of the concentration-response curve in GraphPad Prism 6 software. In assays to evaluate the effect of the test compound alone, membranes were treated with serial dilutions of the compound and were subjected to the same operations as noted above. In functional reversibility assays, serial dilutions of the test compound and serial dilutions of FTY720-P (Cayman) were co-incubated with membranes followed by the same operations as noted above. The fluorescence ratio obtained from the response to 10 μM FTY720-P alone was calculated as a 100% response ($E_{max} = 100\%$), and that in the condition without FTY720-P or the test compound was defined as 0%. The maximum value of the response to FTY720-P in co-treatment with the test compound (calculated by nonlinear regression analysis of the concentration-response curve in GraphPad Prism 6) was regarded as the $E_{max}$ value.

### Receptor internalization assay

HEK293 cells (RIKEN BRC) transiently transfected with pBiT3.1-N in which the coding sequence of human S1PR1 was inserted (pBiT3.1-N-WT-S1PR1), resulting in the expression of S1PR1 with an HiBiT-tag at the N-terminus, were prepared using Lipofectamine 2000 (Thermo Fisher Scientific). In the functional reversibility assay, serial dilutions of the test compound were preincubated with cells in a culture medium consisting of Dulbecco's Modified Eagle's Medium (Wako) supplemented with 10% fetal bovine serum (Biological Industries) for 10 min at 37 °C, followed by the induction of receptor internalization by the addition of serial dilutions of FTY720-P or S1P (Cayman). Following 60-min incubation at 37 °C, cell surface levels of HiBiT-tagged human S1PR1 were monitored using a Nano Glo HiBiT Extracellular Detection System (Promega) according to the manufacturer's protocol. The surface level of receptors, represented by the luminescence, was read by using a PHERAstar FSX microplate reader. The luminesce induced by 10 μM FTY720-P or 100 μM S1P alone was calculated as 100% response ($E_{max} = 100\%$), and that in the condition without agonist or the test compound was defined as 0%. The maximum value of the response to FTY720-P or S1P following pretreatment of cells with the test compound (calculated by nonlinear regression analysis of the concentration-response curve in GraphPad Prism 6) was regarded as the $E_{max}$ value. In functional reversibility assays with cells expressing mutant S1PR1, pBiT3.1-N with insertion of the mutated human S1PR1 sequence where the original codon for Met124 (ATG) was replaced by GTG or CTG, leading to the expression of M124V (pBiT3.1-N-M124V-S1PR1) or M124V (pBiT3.1-N-M124L-S1PR1), respectively, was transfected into cells. In assays to evaluate the effect of the test compound alone, cells in culture medium were treated with the compound for 60 min at 37 °C and the surface S1PR1 levels were evaluated as noted above. In assays to evaluate the antagonistic effect of the test compound, cells in the culture medium were treated with serial dilutions of the compound for 10 min at 37 °C, followed by the addition of 10 nM FTY720-P.

### $Ca^{2+}$ mobilization assay

CHO-K1 cells (ECACC) transiently transfected with human S1PR2 or S1PR3, or cells transiently co-transfected with human S1PR1, S1PR4, S1PR5, mouse S1PR1, or rat S1PR1 and human Gα15, were prepared using FuGENE HD (Promega). $Ca^{2+}$ mobilization assay using a Fluo4 Direct Calcium Assay Kit (Thermo Fisher Scientific) was carried out according to the manufacturer's protocol. Following the pretreatment of Fluo4-loaded cells with serial dilutions of the test compound in an assay buffer consisting of Hanks' Balanced Salt Solution (Thermo Fisher Scientific) and 20 mM HEPES (Thermo Fisher Scientific) supplemented with 0.1% bovine serum albumin (Sigma-Aldrich) and 2.5 mM probenecid (Thermo Fisher Scientific) for 15 min, $Ca^{2+}$ mobilization was induced by 10 nM Merck S1PR1 agonist for cells expressing S1PR1, or by 100 nM S1P for human S1PR2-, S1PR3-, S1PR4-, or S1PR5-transfected cells. The increase in cellular calcium level was evaluated using an FDSS 7000 plate reader (Hamamatsu Photonics) and the maximum fluorescent signal in each well was calculated. $IC_{50}$ values were calculated by nonlinear regression analysis of the concentration-response curve in GraphPad Prism 6 software. In the evaluation of the dissociation half-life of the test compounds from S1PR1, the compound was incubated with cells expressing human S1PR1 and Gα15 in the culture medium consisting of Ham's F-12 Nutrient Mixture (Thermo Fisher Scientific) supplemented with 10% fetal bovine serum for 60 min at 37 °C, followed by the extensive wash with phosphate-buffered saline (Wako). Following incubation for 1–24 h in a culture medium at 37 °C, a $Ca^{2+}$ mobilization assay was carried out as noted above. $Ca^{2+}$ mobilization was induced by a 10 nM Merck S1PR1 agonist. The ratio of the maximum fluorescent signal to the minimum after the addition of agonist in each well was calculated. The ratio obtained from the response to 10 nM Merck S1PR1 agonist was calculated as 100% and that in the condition without the agonist or the test compound was defined as 0% to evaluate the response by receptors spared from compound binding after washout. The dissociation half-life was calculated by nonlinear regression analysis (one phase exponential decay) of the incubation time–response curve in GraphPad Prism 6 software.

### cAMP assay

HEK293 cells transiently transfected with pBiT3.1-N-WT-S1PR1 or pBiT3.1-N-M124V-S1PR1 were prepared using Lipofectamine 2000. Cells were pretreated with serial dilutions of the test compound in assay buffer consisting of Hanks' Balanced Salt Solution, 20 mM HEPES, and 0.5 mM 3-isobutyl-1-methylxanthine (Wako) for 10 min, followed by the addition of serial dilutions of FTY720-P. Following incubation for 10 min, 5 μM forskolin (Wako) was added for 20 min. Cellular cAMP levels were examined using a LANCE Ultra cAMP Kit (PerkinElmer) according to the manufacturer's protocol, and the time-resolved fluorescence resonance energy transfer (TR-FRET) emission (665 nm) was read by using a PHERAstar FSX microplate reader. All procedures were performed at room temperature. The TR-FRET fluorescence obtained from the response to 1 μM FTY720-P alone was calculated as a 100% response ($E_{max} = 100\%$) and that in the condition without FTY720-P or the test compound was defined as 0%. The highest value of the response to FTY720-P in cells co-treated with the test compound (calculated by nonlinear regression analysis of the concentration-response curve in GraphPad Prism 6) was regarded as the $E_{max}$ value.

### Effect on heart rate in guinea pigs

Six-to-seven-week-old male Hartley guinea pigs anesthetized by intraperitoneal injection of 35 mg/kg pentobarbital sodium (Kyoritsu Seiyaku) and 2.5 mg/kg of butorphanol tartrate (Meiji Seika Pharma) were subjected to the insertion of an endotracheal tube via a tracheotomy. The thorax was incised, and the right common carotid artery was catheterized with a PE90 catheter (BD) filled with heparinized saline (50 IU/mL) connected to a blood pressure amplifier (Nihon Kohden) and heart rate counter (Nihon Kohden) via a DTXPlus Transducer (Merit Medical). Signals were monitored and recorded by using a PowerLab 8/35 instrument (AD Instruments) and LabChart

software (AD Instruments). KSI-6666 (30 mg/kg) dissolved in dimethyl sulfoxide was intravenously administered to the animals, followed by an intravenous injection of 0.3 mg/kg fingolimod hydrochloride (LC Laboratories) dissolved in dimethyl sulfoxide 15 min after the administration of KSI-6666. The heart rate was recorded continuously during the study. The average of the heart rate counted for 1 min just before the administration of KSI-6666 (-15 min), just before and every 10 min after the administration of fingolimod (0, 10, 20, and 30 min) was calculated and used for data analysis.

### Chemotaxis assay

Twelve-week-old female BALB/cA mice were sacrificed, and spleens were collected. Splenocytes were prepared after the removal of red blood cells using RBC Lysis Buffer (BioLegend). Cells ($2 \times 10^7$ cells/mL) in assay medium consisting of phenol red-free RPMI-1640 (Thermo Fisher Scientific) supplemented with 0.1% fatty acid-free bovine serum albumin (Sigma-Aldrich) were incubated with serial dilutions of the test compound for 5 min, and 100 μL cell suspensions were transferred to Transwell inserts (Corning) loaded in 24-well culture plates (BD) filled with 600 μL assay medium per well and 20 nM S1P. Cells were incubated for 3 h at 37 °C in a 5% $CO_2$ incubator, followed by removal of the inserts. Migrated cells in plates were stained with CyQuant (Thermo Fisher Scientific) for 1 h at 37 °C in the $CO_2$ incubator and fluorescence intensity, correlated with cell number, was measured by using a PHERAstar FSX microplate reader.

### Naïve CD4⁺CD45RBʰⁱ T cell adoptive transfer colitis model

Eight-to-nine-week-old female BALB/cA mice were sacrificed, and spleens were collected. Splenocytes were prepared after the removal of red blood cells using RBC Lysis Buffer. Naïve T cells were isolated by negative selection using a Mouse Naive T Cell CD4⁺/CD62L⁺/CD44ˡᵒʷ Column Kit (R&D Systems). Naïve CD4⁺ T cells were treated with fluorescein isothiocyanate-labeled (FITC) anti-mouse CD45RB antibody (BioLegend 103305), followed by washing. Cells treated with anti-FITC MicroBeads (Miltenyi Biotec) were magnetically separated via a MACS column (Miltenyi Biotec) and CD4⁺CD45RBʰⁱᵍʰ T cells were thus purified. Cell suspension ($6 \times 10^5$ cell/mL) was intraperitoneally transferred into 8–10-week-old female C.B-17 SCID mice (500 μL/mouse). Two weeks after the transfer, oral administration of compounds suspended in the vehicle consisting of a 1:9 mixture of 0.5% methylcellulose 400 solution and distilled water twice daily was started therapeutically and lasted for 14–16 days. An equimolar amount of hydrochloric acid was mixed only in the suspension of KSI-6666. The day after the last administration, the DAI score, consisting of the sum (0–12 points) of the grades for three observations [weight loss (%; none: 0; 0–5: 1; 5–10: 2; 10–20: 3; >20: 4); stool consistency (normal: 0; loose stool: 2; diarrhea: 4); and hematochezia (absence: 0; presence: 4)] was evaluated. Mice were sacrificed and colons were collected, weighed, and further evaluations were performed. The histopathological assessment was performed using a modification of the previously described method[50]. The following scoring system, consisting of the sum (0–30 points) of the grades for several observations [decrease in goblet cells (0, 2, 4, 6); inflammatory cell infiltration (0, 2, 4, 6); mucosal thickening (0, 2, 4, 6); crypt score based on abnormal crypt architecture including distortion, branching, atrophy, and crypt loss (0, 3, 6, 9); and mucosal erosion and ulceration (0, 3)], were assessed by a veterinary pathologist in no double-blinded fashion. For mRNA expression analysis, total RNA was extracted from colons by using an RNeasy Mini Kit (QIAGEN). cDNA was reverse transcribed by using a PrimeScript RT Reagent Kit Perfect Real Time (TaKaRa Bio Inc.). Real-time PCR was performed using a 7500 Fast Real-Time PCR System (Applied Biosystems). SYBR Premix Ex Taq Perfect Real Time (TaKaRa Bio Inc.) for S100a9 with a primer set (MA058882) obtained from TaKaRa Bio Inc. and Premix Ex Taq Perfect Real Time (TaKaRa Bio Inc.) for 18S rRNA with TaqMan Ribosomal RNA Control Reagents (Applied

Biosystems) were used for the reaction. The ΔΔCt method was applied to calculate the relative expression level of the target mRNA using the 18S rRNA gene as the reference. % increase in body weight was calculated by the body weight ratio between the day compound administration started and the end of the study.

### Oxazolone-induced colitis model

A total of 150 μL of 3% 4-ethoxymethylene-2-phenyl-2-oxazolin-5-one (oxazolone; Sigma-Aldrich) dissolved in 100% ethanol (Wako) was applied on shaved back skins of 9–10-week-old male BALB/cCr mice for sensitization. One week after sensitization, 100 μL of 1% oxazolone dissolved in 50% ethanol oxazolone was intrarectally administered to the mice. On the same day, test compounds suspended in the vehicle consisting of a 1:9 mixture of 0.5% methylcellulose 400 solution and distilled water were orally administered to mice twice (before and after sensitization). Mice were sacrificed 24 h after sensitization, and colons were collected and weighed. Homogenates of colons were prepared using TisseLyser (QIAGEN) and the amount of interleukin (IL)-4 protein was evaluated using a Mouse IL-4 Quantikine ELISA Kit (R&D Systems). Change in body weight was calculated by the difference between the day of intrarectal oxazolone administration and the end of the study.

### Experimental autoimmune encephalomyelitis (EAE) in rats

The immunization emulsion consisted of syngeneic CNS antigen (one part brain to two parts of spinal cord) in phosphate-buffered saline (Thermo Fisher Scientific) emulsified 1:1 in complete Freund's adjuvant (Thermo Fisher Scientific) was intradermally injected into two different sites on the dorsal base of the tail root of the 9-week-old female DA rat under anesthesia at 100 μL/site. On the next day of the immunization, oral administration of compounds suspended in the vehicle consisting of a 1:9 mixture of 0.5% methylcellulose 400 solution and distilled water twice daily was started prophylactically and lasted for 12 days. An equimolar amount of hydrochloric acid was mixed only in the suspension of KSI-6666. Rats were daily scored by the following criteria [0: normal appearance; 1: limp tail; 2: unilateral partial paralysis of hind limbs or front limbs; 3: bilateral complete paralysis of hind limbs or front limbs; 4: quadriplegia; 5: death] during the study period started from the day of immunization (day 0) and lasted until day 13. The area under curve (AUC) of the score over time was calculated by GraphPad Prism 6.

### Experimental autoimmune encephalomyelitis (EAE) in mice

EAE was induced in 9-week-old female C57BL/6JJcl mice using an EAE induction kit (Hooke Laboratories) according to the manufacturer's protocol. Briefly, anesthetized mice were immunized by the subcutaneous injection of myelin oligodendrocyte glycoprotein peptide (MOG$_{35–55}$) at two different sites of the back (100 μL/site), followed by the intraperitoneal injection of 100 ng pertussis toxin (day 0). The pertussis toxin injection was repeated on the following day (day 1). Mice were daily weighed and scored by the following criteria [0: normal appearance; 0.5: weak tail; 1: limp tail; 1.5: limp tail + weak hind limbs; 2: unilateral partial paralysis of hind limbs; 2.5: unilateral complete paralysis of hind limbs or bilateral partial paralysis of hind limbs; 3: bilateral complete paralysis of hind limbs; 3.5: bilateral complete paralysis of hind limbs + hind limbs clustered on one side of the body; 4: bilateral complete paralysis of hind limbs + partial front limb paralysis; 5: moribundity or death] on day 2, day 4 and daily basis from day 6. The day after the first observation of clinical score 1 or above in each animal, oral administration of compounds dissolved in the vehicle, which consisted of 5% dimethyl sulfoxide and 5% ethanol mixed in PEG300 (Hampton Research), once daily was started therapeutically and lasted for 14 days. The final assessment was performed on the day after the final administration. The clinical score and body weight of the animal resulting in the death was recorded as 5 and the weight before death, respectively, for the remaining days of the experiment. Animals weighing less than 17 g on day 2 were excluded from the study.

## Molecular docking analysis

Our approach to docking the (R)-KSI-6666 molecule to the potential binding sites in S1PR1 used two steps: (1) structure and ligand preparation for docking, and (2) docking of the (R)-KSI-6666 molecule. An initial structure of human S1PR1 was modeled from the X-ray structure of S1PR1 with the W146 molecule excluding the Escherichia virus T4 structure (PDB: 3V2Y)[35] and refined for docking simulations using the Protein Preparation Wizard Script within Maestro[51] (Schrödinger LLC, New York, NY, USA). For the (R)-KSI-6666 molecule, ionization and energy minimization were performed by the OPLS3 force field in the LigPrep Script in Maestro. A minimized structure of (R)-KSI-6666 was employed as the input structure for docking simulation. To account for both ligand and receptor flexibility in the first step, the Glide 'Induced Fit Docking (IFD)' protocol[52] (Schrödinger LLC) was used, followed by iteratively combining rigid receptor docking using Glide and protein remodeling by side chain searching and minimization using Prime (Schrödinger LLC) techniques. We generated 20 initial orientations of (R)-KSI-6666 in a grid box defined by the center of the co-crystallized ligand (W146) using Glide docking [standard precision (SP) mode]. Next, we applied the soften-potential docking options, which involved scaling the van der Waals radii by 0.5 for receptor and (R)-KSI-6666 atoms. In the protein remodeling stage, all residues within a 5.0-Å radius of each initial docked (R)-KSI-6666 were refined using Prime. (R)-KSI-6666 was then redocked into the refined receptor structure using Glide in SP mode. Finally, the best pose for (R)-KSI-6666 was rescored according to GlideScore (SP mode). The Docking score of co-crystallized ligand W146 was calculated using the score-in-place procedure of Glide SP mode.

## Conventional molecular dynamics (cMD) and metadynamics (MetaD) simulation

Two types of MD simulations were performed. One is a cMD simulation to evaluate the stability of the docking pose and the other is a MetaD simulation to evaluate the ligand-binding persistency. Binding models of (R)-KSI-6666 and W146 to the structure of human S1PR1 were placed in a large palmitoyl-oleoyl-phosphatidylcholine (POPC) bilayer, and transferable intermolecular potential 3 point (TIP3P) water molecules were solvated with 0.15 M NaCl. After minimization and relaxation of the model, the productions of cMD and MetaD phases were performed for five independent 500-ns and ten independent 100-ns simulations, respectively, using Desmond version 2.3 (Schrödinger LLC) in the isothermal–isobaric (NPT) ensemble at 300 K and 1 bar using a Langevin thermostat. The OPLS3e force field[53,54] was used for the simulations. The long-range electrostatic interactions were computed using the Smooth Particle Mesh Ewald method. All system setups were performed using Maestro. The MM-GBSA protocol (Schrödinger LLC) was used to calculate the binding free energy of ligands based on cMD trajectories, and the last 100 ns of 500-ns trajectory each was extracted and summed for the analysis. MetaD simulation is a widely used enhanced sampling method that allows the sampling of free-energy landscapes. In this simulation, we defined the biasing collective variables as the distance between the center of mass of the ligands and the ligand-binding residues (Tyr29, Lys34, Tyr98, Asn101, Ser105, Gly106, Thr109, Trp117, Arg120, Glu121, Met124, Phe125, Leu128, Ser192, Val194, Leu195, Pro196, Tyr198, Ile203, Cys206, Thr207, Phe210, Trp269, Leu272, Phe273, Leu276, Glu294, Leu297, Val298, Ala300 and Val301 for (R)-KSI-6666; Tyr29, Lys34, Tyr98, Asn101, Ser105, Gly106, Thr109, Trp117, Arg120, Glu121, Met124, Phe125, Leu128, Thr193, Val194, Leu195, Pro196, Ile203, Cys206, Thr207, Phe210, Trp269, Leu272, Phe273, Leu276, Glu294, Leu297, Ala300, and Val301 for W146). The initial Gaussian hill height was set at 0.3. The MetaD procedure for (R)-KSI-6666 in the M124V variant was identical to that for wild-type S1PR1. The model of M124V-S1PR1 was constructed using Maestro.

## Protein–ligand interaction fingerprint analysis

PLIF analyses from MetaD trajectories were performed using the PLIF modules of MOE 2020.0901 (Chemical Computing Group Inc., Montreal, Quebec, Canada). Energy-based contacts, including side chain hydrogen bonds (donor or acceptor), backbone hydrogen bonds (donor or acceptor), solvent hydrogen bonds (donor or acceptor), ionic interactions, metal binding interactions, and π interactions, and surface contact fingerprints, were employed. After the PLIFs had been generated based on 1000 snapshot structures (every 1 ns in ten independent 100-ns simulations) for each ligand, the probability of each bit occurring for each residue for (R)-KSI-6666 compared with W146 was calculated as Qb.

$$Q_b = \frac{C_b}{C_0} \cdot 2\left(P_b - \frac{1}{2}\right)$$

Cb is the number of snapshots within the set that have the fingerprint bit set, and C0 is the total number of snapshots. Pb is the relative probability of the fingerprint bit entry over the snapshots for (R)-KSI-6666 compared with that for W146 if it contains the bit. Qb takes values from −1 to 1. A higher positive value indicates a higher probability of the bit occurring for (R)-KSI-6666.

## Quantification and statistical analysis

Graphs are presented as mean ± s.e.m, as indicated in the figure legends. Three or more independent experiments were performed in each in vitro study. Three or more animals/groups were used in each in vivo study. Statistical analyses were performed by F-test, Student's t-test (two-tailed), Welch's t-test (two-tailed), one-way ANOVA, two-way ANOVA, Dunnett's test, and Tukey's test. p values and sample sizes can be found in the figure legends. No randomization or blinding was used.

## Reporting summary

Further information on research design is available in the Nature Portfolio Reporting Summary linked to this article.

## Data availability

Source data are provided within this paper. The MD simulation data including input files and final output configurations have been deposited to GitHub depository, https://github.com/CHEMINFO-tsukuba/Nat.Comm.2024.git. Photo image data generated in the current study are available in the Zenodo database, https://doi.org/10.5281/zenodo.11260928. Source data are provided with this paper.

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

## Acknowledgements

This work was supported by the Research Support Project for Life Science and Drug Discovery (Basis for Supporting Innovative Drug Discovery and Life Science Research [BINDS]) from AMED (grant number JP22ama121029j0001 to T.H.), Grants-in-Aid for Scientific Research from JSPS (17H04038, 17K08622, 20K07332, and 20H03441 to T.A. and N.A.), Grants-in-Aid for Scientific Research in Innovative Areas from MEXT (18H04989 and 19H04821 to T.A. and N.A.) and by CREST from the Japan Science and Technology Agency (JPMJCR2011 to T.A.). We thank Edanz (https://jp.edanz.com/ac) for editing a draft of this manuscript.

## Author contributions

Y.M., Y. Ohsawa, T.S., M.M., T.M., N.A., W.M., K.N. and N.H. performed the experiments. K.O. and H.I. synthesized the chemicals. A.Y. performed LC-MS analysis. M.H. and Y. Okuhara performed histological analysis. Y.Y. and T.H. performed in silico calculations. Y.M., Y.Y., T.I., K.H., M.H., T.H. and T.A. analyzed the MetaD data. Y.M. and T.A. designed the study and wrote the paper.

## Competing interests

YM, Y. Ohsawa, TS, YY, KO, HI, AY, MH, and Y. Okuhara are employees of Kissei Pharmaceutical Co., Ltd. and may hold stock in the company. The remaining authors declare no competing interests.
