## [Peer Review File · Nature Communications]

Pseudoirreversible inhibition elicits persistent efficacy of a sphingosine-1-phosphate receptor-1 antagonistReviewers' comments:

Reviewer #1 (Remarks to the Author):

In this manuscript, the authors described a new S1PR1 inhibitor - KSI-6666, which seems to be a “pseudoirreversible competitive” inhibitor for this pharmacologically important GPCR. The authors show by standard GPCR assays, in vivo efficacy assays and molecular dynamic simulation studies that the unique structure of this compound likely interacts with unique structural elements of S1PR1 in a pseudoirreversible manner – i.e. poor dissociation of receptor-drug complex which likely explains the in vivo properties of this compound in lymphocyte reduction kinetics.

The receptor signaling studies are generally well done. The data document the novel properties of the novel compound KSI-6666. Due to the potent lymphopenic effects coupled with lack of bradycardia side effect, this compound is potentially interesting from a clinical perspective. However, as one would expect, this compound induces lung vascular permeability by inhibiting endothelial S1PR1. This effect would limit its utility as an anti-inflammatory and/or immunomodulatory drug in autoimmune indications. This is the most important issue that the authors have not addressed. If this compound works as well as S1PR1 functional antagonists in the EAE or other autoimmune models, it would suggest potential utility. This important issue should be addressed by including new data.

Figure 3g-h contains studies to determine if KSI-6666 receptor interaction can be reversed by FTY720p or S1P. This monitored receptor internalization. It would be helpful to show representative images from this assay.

Overall, this manuscript contains studies with an interesting compound KSI-6666 which seems to inhibit S1PR1 in a unique way and possesses interesting pharmacological properties. However, the authors have not addressed whether the significant adverse event of lung vascular leak would prevent the potential utility of this agent as a new therapeutic paradigm. If this was addressed and found to be useful, this work will have significant impact.

Reviewer #2 (Remarks to the Author):

The article „Pseudoirreversible inhibition elicits persistent efficacy of a S1P1R antagonist” by Maruyama et al., introduces a potent, and in vivo efficacious molecule KSI-6666.

The in vivo experiments and the long duration of action in vivo is very impressive, but the connection between the long residence time of KSI-6666 and its action cannot be demonstrated by the experiments provided in a sound and convincing manner. In order to make such a connection, there are several experiments missing in the article, and especially on the modeling part, conclusions do not stem from state-of-the-art approaches.

Major points:

- The authors demonstrate the long duration of action in vivo, but they do not corroborate the pharmacokinetics of KSI-6666. In Figure 7 they report the plasma concentrations after several hours, but they do not discuss the pharmacokinetics/pharmacodynamics relationship. From the data I can see here, I would suggest, that the long duration of action in vivo comes from a long plasma residence time rather than a slow off-rate. This misconception by the authors is underlined by their statement in line 384 on Aprepitant – if they check the underlying data (e.g., by reading the Dahl & Akerud paper 2013, that is cited), they will see that the duration of action is only caused by plasma residence time rather by receptor residence time (the latter being factor 19 shorter). Also in the case of KSI-6666 the plasma residence time seems to be quite long.

- The study concentrates on the long residence time of KSI-6666, but the residence time is never determined. For me it would be mandatory to determine the off rate experimentally to substantiate the claims. For example, in a study by Sykes et al. in 2014 the off rates of fingolimod and S1P have been determined, so there is precedence.

- The assay incubation times are non-adequate for ligands with a long residence time. As a rule of thumb, the test compounds need to incubate with the receptor 3 times the half life (which is admittedly not known). I strongly recommend repeating the assays with significantly longer incubation times to confirm the reduction of maximal activation.

- Modeling procedures:

o The authors decided to take an inactive state Xray structure (3V2Y) as starting point and docked KSI-6666 in a quite elaborate fashion. As a matter of fact, they ignored that cryoEM structures with exactly the same the zwitterionic head group as KSI-6666 are available (e.g., 7EO4 – Siponimod bound S1P1R), which would be a good guidance for the location of the headgroup. I am fully aware that 7EO4 is an active structure, but this does not mean that the location of headgroup will be different – especially because the molecules differ in the activation relevant TM6 region.

o Especially the fact, that the metaMD simulations show a minimum 2 Angstrom from the docking pose is a clear hint, that the docking pose may not be accurate. The authors should check the validity of the docking pose by plain MD simulations (several 100ns) to check its stability. Beyond that – as the authors use the Schrödinger software, they should use the docking pose to do FEP

calculations on the KSI compound and the analogs – if the pose was correct, the plain MD should reveal a stable binding mode and the FEP calculations should be able to rank the analogs.

o Connection between metaMD results and ligand residence time – there the authors do not provide any real scientific reasoning. They have not used the procedure by Tivary to compute residence times, but they discuss different binding pockets. That may be a correct observation, but to me it is not clear how this relates to residence time. The plots in Figure 5 do not give too much insight – it would be interesting to see, where in the receptor these pockets are exactly located – especially for S1P receptors, the ligand entry/exit channel is close to the membrane and I expect membrane involvement as well ...

o The authors should also show the exit paths of the ligands by metaMD – they just define the binding site residues of the ligands and deliberately define the c.o.m. of the ligands and the binding site as collective variable. Why did they do so and what were the results (geometrically – along the exit channel of the receptor)?

o The FMO calculations are done on snapshots of the metaMD simulation – most importantly I'd like to see a stable binding mode and a FMO calculation on this. In addition, I do not understand how the FMO calculations can help to solidify the claim of slower off-rates. If the authors would claim stronger binding for different decorations of the phenyl ring, this would be in scope of the theory, but kinetics are clearly not the goal of a FMO calculation

Minor:

- Caption Figure 3d: The term is “exchange energy” rather than “exclusion energy”

Taking all this together, I can clearly not recommend the publication of this manuscript.

Reviewer #3 (Remarks to the Author):

Overall, the article provides detailed insights into the development and characterization of KSI-6666 as a selective antagonist of S1PR1. The compound shows promising potential for the treatment of intestinal inflammation and offers advantages such as persistent efficacy and pseudoirreversible inhibition.

The study is significant to the field and related fields and meets all the expected standards for experimental work. The article highlights the discovery of KSI-6666 as a new pseudoirreversible inhibitor of S1PR1, which exhibited persistent efficacy *in vivo*. The study also proposes the two-step induced fit model to explain the slow dissociation of KSI-6666 from S1PR1. The role of Met124 in the binding pocket formation and the importance of the benzene ring moiety in KSI-6666 are also noteworthy findings.

A comprehensive methodological approach was used in the study, including molecular dynamics and quantum mechanics methods, along with experimental validation. Methods are well described with enough details for the work to be reproduced.

The data are well analysed and properly presented. Presented data, along with the computational predictions and experimental validations, support the conclusions and claims made regarding the pseudoirreversible inhibition of S1PR1 by KSI-6666 and the role of Met124 in the binding process.

However, there is a lack of comprehensive comparison. The discussion part is poor. The article does not explicitly discuss or compare its findings with the existing literature on S1PR1 inhibition. It would be beneficial to provide a thorough review and comparison of previous studies, highlighting the novelty and significance of the current findings. Also, possible limitations of the study should be identified.

Reviewer #4 (Remarks to the Author):

The manuscript by Maruyama Y et al., aims to find a potent inhibitor of sphingosine 1-phosphate receptor (S1PR1) for treatment of inflammatory disorders. A lead molecule is used to screen candidate inhibitors in vitro and in vivo, to determine the impact on S1PR1 signalling and lymphopenia. The newly identified compound KSI-6666 is then tested in two pre-clinical murine models of inflammatory bowels disease. Administration of the compound significantly reduced inflammation in both models in a similar manner to a known S1PR1 inhibitor, KRP. Molecule modelling of the interaction of KSI-6666 with S1PR1 was then carried out to determine mechanism of inhibition (requirement of a binding of benzene ring KSI-6666 within a methionine in the ligand-binding pocket of S1PR1).

Previous publications have shown that S1PR1 regulated cell migration during inflammation. Several inhibitors for S1PR1 exist, including FTY720, a small molecule that is phosphorylated in vivo to form FTY720-P, binds to S1PR and mediates their internalization and degradation. Although effective in pre-clinical models of renal transplant, this drug elicited a transient, asymptomatic bradycardia in patients (Budde K et al., 2002). This led to the development of novel inhibitors of S1PR1, including molecules with a similar structure to FTY720-P, for example, W146, which had a poor efficacy in vivo, or structurally distinct molecules, for example NIBR-0213. Further refinement of drug design may find more effective molecules for treatment of inflammatory disorders.

I have some concerns over the pre-clinical models and some other major points raised below, which require corrections to the current manuscript before it is acceptable for publication within Nature Communications.

Major comments:

1. For the T-cell transfer model, the purity of transferred T cell populations should be shown. The authors state that they treat with the compounds for 14-16 days two weeks after transfer, a total of 4-weeks for the model. According to published protocols, colitis in this model is readily detectable from 5 weeks, with a weight loss becoming evident 3-5 weeks post-reconstitution (Powrie F et al., 1993). Because of this difference in time-course and the variability of the DAI score given in Figure 2A, it would be useful to see weight loss as an independent Figure. In addition, histological score is often comprised of more extensive parameters, including abnormal crypts (score 0–3), presence of crypt abscesses (score 0–1), mucosal erosion and ulceration (score 0–1) and submucosal spread to transmural involvement (score 0–3) (Ostanin D et al., 2009). To support their Figure, the authors should provide several clear images with the histological features highlighted for each treatment group. Authors also should include a statement regarding whether section scoring was blinded and whom this was performed by. Colon weight in Figure 2a should be re-labelled colon length to weight ratio (ratios for animals with colitis are usually between 0.02 and 0.12 g/cm). A precedent for using S1009 mRNA/18S rRNA ratio should be established, given this marker is reported to be upregulated during chronic colitis (Brudzewsky D et al., 2009).

2. For the oxazalone-induced colitis model, it is more common to see colon length reported as a sign of inflammation and colon shortening, not colon weight (Figure 2C). Weight loss of animals would be useful to see here. Was the colon tissue stimulated before measuring IL-4 protein production? IL-4 production is variable in this model (see Boirivant M et al., 1998). Some examples of gross pathology or histology would support conclusions on inhibition of inflammation from the Figures shown.

3. It is not clear why lymphocyte count is shown for cynomolgus monkeys (Figure 1J), with leukocyte count for rats (Figure 1B) and heart rate for guinea pigs (Figure 1G). Some consistency in recordings for both parameters in rodent and primate species is needed. Details of the guinea pigs should be given in the Materials and Methods.

4. Given the variable effect of KSI, particularly the significance in inhibition of inflammation compared to KRP, it is not possible to conclude that the authors have achieved their aim of discovering a novel potent antagonist for the treatment of inflammatory disease.

5. Clear details of how compound 1 was identified as a lead compound need to be given. This sets the scene for identification of KSI-6666.

6. Given the introduction sets a precedent for the identification of new S1PR1 inhibitors for treatment of autoimmune conditions, such as MS, it is not clear why the authors test their compounds in two differing models of inflammatory bowel disease. Neither of these pre-clinical models represent an autoimmune inflammatory condition.

7. A clearer definition of a “pseudoirreversible inhibitor” needs to be given. From my understanding this term applies to covalent inactivation of an enzymatic active site, followed by spontaneous hydrolysis, which results in restoration of function. It is not clear from the experiments shown that KSI-6666 fulfils these criteria.

8. It is not clear from the experiments shown, how KSI-6666 can be deemed to have “inverse agonistic activity”. From the definition given, this term applies to spontaneous receptor activation, not competitive inhibition of receptor signalling.

Point by point response to reviewer's comments

Thank you for all constructive comments and criticisms. The major changes are highlighted in yellow in the revised manuscript.

Reviewer #1 (Remarks to the Author):

In this manuscript, the authors described a new SIPR1 inhibitor - KSI-6666, which seems to be a “pseudoirreversible competitive” inhibitor for this pharmacologically important GPCR. The authors show by standard GPCR assays, in vivo efficacy assays and molecular dynamic simulation studies that the unique structure of this compound likely interacts with unique structural elements of SIPR1 in a pseudoirreversible manner – i.e. poor dissociation of receptor-drug complex which likely explains the in vivo properties of this compound in lymphocyte reduction kinetics.

The receptor signaling studies are generally well done. The data document the novel properties of the novel compound KSI-6666. Due to the potent lymphopenic effects coupled with lack of bradycardia side effect, this compound is potentially interesting from a clinical perspective. However, as one would expect, this compound induces lung vascular permeability by inhibiting endothelial SIPR1. This effect would limit its utility as an anti-inflammatory and/or immunomodulatory drug in autoimmune indications. This is the most important issue that the authors have not addressed. If this compound works as well as SIPR1 functional antagonists in the EAE or other autoimmune models, it would suggest potential utility. This important issue should be addressed by including new data.

We performed experiments of rodent EAE models to evaluate the efficacy of KSI-6666 on autoimmune disease. Data suggested that KSI-6666 suppresses the disease onset and progression in these models. We exhibit these data in Fig. 2a and b.

Figure 3g-h contains studies to determine if KSI-6666 receptor interaction can be reversed by FTY720p or SIP. This monitored receptor internalization. It would be helpful to show representative images from this assay.

> Unfortunately, it is difficult to have a typical image for the receptor internalization in the current assay. Instead, we confirmed the receptor internalization caused by FTY720-P and suppression of the internalization by the pre-treatment of KSI-6666 in flow cytometric analysis. So, the flow cytometric profiles are exhibited in Extended Data Fig. 4a of the revised manuscript.

Overall, this manuscript contains studies with an interesting compound KSI-6666 which seems to inhibit S1PR1 in a unique way and possesses interesting pharmacological properties. However, the authors have not addressed whether the significant adverse event of lung vascular leak would prevent the potential utility of this agent as a new therapeutic paradigm. If this was addressed and found to be useful, this work will have significant impact.

> When KSI-6666 was administered, it caused pulmonary vascular leakage, a response similar to that observed with fingolimod, as exhibited in Extended Data Fig. 2a and b. Unfortunately, this unfavorable effect appears to occur commonly in any S1PR1 modulators, including those approved for clinical use. However, additional experiments revealed that the administration of KSI-6666 did not noticeably affect lung function parameters such as Penh and f. Together with no incidence of bradycardia by the KSI-6666 treatment, these data suggest the potential benefit of using KSI-6666. We will add these data to Extended Data Fig. 2c to the manuscript.

Reviewer #2 (Remarks to the Author):

The article „Pseudoirreversible inhibition elicits persistent efficacy of a SIP1R antagonist” by Maruyama et al., introduces a potent, and in vivo efficacious molecule KSI-6666. The in vivo experiments and the long duration of action in vivo is very impressive, but the connection between the long residence time of KSI-6666 and its action cannot be demonstrated by the experiments provided in a sound and convincing manner. In order to make such a connection, there are several experiments missing in the article, and especially on the modeling part, conclusions do not stem from state-of-the-art approaches.

Major points:

- The authors demonstrate the long duration of action in vivo, but they do not corroborate the pharmacokinetics of KSI-6666. In Figure 7 they report the plasma concentrations after several hours, but they do not discuss the pharmacokinetics/pharmacodynamics relationship. From the data I can see here, I would suggest, that the long duration of action in vivo comes from a long plasma residence time rather than a slow off-rate. This misconception by the authors is underlined by their statement in line 384 on Aprepitant – if they check the underlying data (e.g., by reading the Dahl & Akerud paper 2013, that is cited), they will see that the duration of action is only caused by plasma residence time rather by receptor residence time (the latter being factor 19 shorter). Also in the case of KSI-6666 the plasma residence time seems to be quite long.

> Thank you for your critical comment. We have acquired pharmacokinetic data indicating a half-life of 6.7 hours for the elimination of KSI-6666 from the blood (Figure 4e). Moreover, we have determined the dissociation half-life of KSI-6666 to be 9.4 hours (Figure 4d). These data suggested that the dissociation half-life appears to be longer than the plasma half-life. As pointed out by reviewer, Dahl et al. proposed that prolongation of binding owing to a long drug–target residence time can occur when the binding dissociation is slower than the pharmacokinetics elimination (Dahl & Akerud paper 2013). If their criteria are applied, KSI-6666 is classified as a drug of which persistency comes from a long drug-target residence time. However, it is also worth noting that the pharmacokinetic rate of the KSI-6666 elimination from the blood is slow, which may also contribute to its persistent effects. Taking these factors into consideration, we have included a discussion on this point in our revised manuscript (Page 16, line 443). In this discussion, we omitted the description regarding aprepitant.

- The study concentrates on the long residence time of KSI-6666, but the residence time is

never determined. For me it would be mandatory to determine the off rate experimentally to substantiate the claims. For example, in a study by Sykes et al. in 2014 the off rates of fingolimod and SIP have been determined, so there is precedence.

> Related to the above, an in vitro dissociation rate was determined (Fig. 4d). As demonstrated by Sykes et al., conducting dissociation experiments using ³H-labeled compounds is considered one of the best ways to directly determine the dissociation rate. However, because the synthesis of ³H-labeled KSI-6666 is very costly and time-consuming, we performed an indirect dissociation assay. We estimated the dissociation half-life of KSI-6666 from S1P1R as 9.4 hr. We added these data in Fig. 4e in revised manuscript and describe the result (Page10, line 271).

- The assay incubation times are non-adequate for ligands with a long residence time. As a rule of thumb, the test compounds need to incubate with the receptor 3 times the half life (which is admittedly not known). I strongly recommend repeating the assays with significantly longer incubation times to confirm the reduction of maximal activation.

> Thank you for your comment. We confirmed a reduction in E_{max} after longer incubation times (5 hr and 10 hr) of KSI-6666, as depicted in Extended Data Fig. 4b (Page 10, line 263). Unfortunately, extending the incubation time to above 24 hr was challenging due to the loss of cellular response to FTY720 for unidentified reasons. Consequently, these incubation times fall short of the duration proposed by the reviewer (28 hr, equivalent to 3 times the half-life of 9.4 hr), which is expected to be enough time for KSI-6666 to bind to all S1PR1 receptors.

Due to the considerable difficulty in achieving a 28-hour incubation, we conducted an assessment to determine whether a 10-hour incubation period would adequately address the concerns raised by the reviewer. Initially, we derived a dissociation rate constant of 0.074 hr⁻¹ from a dissociation half-life of 9.4 hr, assuming normal equilibrium of binding and dissociation between KSI-6666 and S1PR1. Considering IC₅₀ of Ca²⁺ mobilization experiments in Fig 1e, the equilibrium dissociation constant should be around 2.5 x 10⁻⁹ M, giving an association kinetics constant above 3.0 x 10⁷ M⁻¹ hr⁻¹. For 1 x 10⁻⁶ M of KSI-6666, the estimated incubation time required for 95% receptor binding is around 6.0 minutes. As a result, similar to the 28-hour incubation, the 10-hour incubation should allow KSI-6666 to occupy almost all of S1PR1. Notably, this estimation also implies that the incubation time of 10 minutes used in this study should be sufficient for KSI-6666 to bind to over 95% of the receptor.

- *Modeling procedures:*

o *The authors decided to take an inactive state Xray structure (3V2Y) as starting point and docked KSI-6666 in a quite elaborate fashion. As a matter of fact, they ignored that cryoEM structures with exactly the same the zwitterionic head group as KSI-6666 are available (e.g., 7EO4 – Siponimod bound S1P1R), which would be a good guidance for the location of the headgroup. I am fully aware that 7EO4 is an active structure, but this does not mean that the location of headgroup will be different – especially because the molecules differ in the activation relevant TM6 region.*

> The reviewer suggests using the structure determined by Cryo-EM, which was released in January 2022, as the initial structure for the docking experiment. Unfortunately, we had already completed most of our MD calculations by the time it was released. Therefore, we have performed the docking experiment using this S1P1R structure (7EO4) and exhibited in Extended Data Fig. 3 and describe this in Page 8, line 209. The visible inspection suggested that the docking structure seems to be reasonably similar with that obtained using 3V2Y.

Yet, we acknowledge that it may provide some information to redo the MetaD calculation based on this structure. Nevertheless, we are unsure if it would yield new findings because of 2 main reasons below.

1) Because of the similar pose of KSI-6666 with that obtained using 3V2Y structure, it is likely that MetaD calculation using 7EO4 structure would yield similar outcomes with our current analysis.

2) As also mentioned by the reviewer, 7EO4 is a siponimod-bound structure with a functional agonist. It would be quite useful to understand the structure activating downstream signaling. On the other hand, KSI-6666 is a competitive inhibitor, and would not change the receptor structure to the active form. Thus, the structure of the receptor when bound to KSI-6666 is likely to be considerably different from that of the siponimod-bound form.

Finally, as compared to MD calculations we performed, it is likely that the conformational change from the active to the inactive form in the MD calculation is necessary for the MD calculation based on 7EO4 structure, which is expected to take more than 10 times for the calculation, which makes it difficult to apply in our current computational environment.

Consequently, while we show the docking pose of KSI-6666 in 7EO4 structure as Extended data Fig. 3, we have not redone MetaD calculation using this docking structure.

o Especially the fact, that the metaMD simulations show a minimum 2 Angstrom from the docking pose is a clear hint, that the docking pose may not be accurate. The authors should check the validity of the docking pose by plain MD simulations (several 100ns) to check its stability. Beyond that – as the authors use the Schrödinger software, they should use the docking pose to do FEP calculations on the KSI compound and the analogs – if the pose was correct, the plain MD should reveal a stable binding mode and the FEP calculations should be able to rank the analogs.

> We conducted the conventional MD simulations of KSI-6666 and control W146 for a total time of 2500 ns (500 ns x 5 times), confirming the stability of the docking pose, and included this result in Figure 3d of the resubmitted manuscript (Page 8, line 205). From this MD calculation, we evaluated the binding free energy of these compounds as -68.3 kcal/mol for KSI-6666 and -65.4 kcal/mol for W146, suggesting that the binding pose of KSI-6666 should be stable as W146, which is the ligand in the original crystal structure. We understand that FEP calculations are a very accurate method of calculation. Unfortunately, however, Schrödinger's FEP license is costly for us and we cannot afford to operate it. Moreover, FEP calculation provides accurate static binding structures, but may not be appropriate for predicting the dissociation process. Accordingly, we have not performed the FEP calculation in this study.

o Connection between metaMD results and ligand residence time – there the authors do not provide any real scientific reasoning. They have not used the procedure by Tiwary to compute residence times, but they discuss different binding pockets. That may be a correct observation, but to me it is not clear how this relates to residence time.

> Thank you for this proposal. Tiwary's method is a powerful technique for calculating dissociation rate constant (*k_{off}*) through metadynamics calculations, and a prerequisite for its implementation is to define appropriate collective variables (CVs) that represent the compound dissociation process. It is a fact that if one wishes to predict the dissociation process of KSI-6666 solely through computational simulation, the application of metadynamics using Tiwary's method will yield significant results. On the other hand, since

KSI-6666 is a new compound, its exact CVs required to interpret the dissociation process was unclear. Therefore, metadynamics calculation of the process toward the fully dissociated state, typically required for the application of the Tiwary method, was not performed, and indeed, some trajectories appear to remain in the undissociated state after 100 ns in our simulations. Consequently, we believe that setting appropriate CVs and calculating reliable *k_{off}* based solely on the current metadynamics data is challenging for KSI-6666.

Even though our selection of CVs, specifically the distance between the centers of gravity of the compound and the distance between the centers of gravity of the amino acid residues around the binding site, in the current metadynamics simulation, was tentative and may not be perfect, it is noteworthy that the uniqueness of this study lies in proposing dissociation processes by aligning predictions from the metadynamics method with in vitro and in vivo experimental data. Accordingly, we think that the analysis of dissociation kinetics using advanced computational simulations such as the Tiwary method is a subject for future study.

The plots in Figure 5 do not give too much insight – it would be interesting to see, where in the receptor these pockets are exactly located – especially for SIP receptors, the ligand entry/exit channel is close to the membrane and I expect membrane involvement as well ...

> We exhibited where these pockets are located in the S1PR1 in Figure 6c and describe its location in the manuscript (Page 13, line 367)

o The authors should also show the exit paths of the ligands by metaMD – they just define the binding site residues of the ligands and deliberately define the c.o.m. of the ligands and the binding site as collective variable. Why did they do so and what were the results (geometrically – along the exit channel of the receptor)?

> The revised manuscript includes snapshots illustrating the exit paths of KSI-6666 from the receptor in Figure 3h, captured during the MetaD simulation. KSI-6666 was observed to be released from the S1PR1 along the path between helices I and VII, aligning with the pathway previously proposed (PMID: 22344443). This finding was described on Page 9, line 244.

Regarding our CV settings, due to a lack of prior knowledge on the dissociation process of the novel compound KSI-6666, we hypothesized that this specific CV configuration would

effectively monitor and provide insights into the dissociation behavior of KSI-6666. Therefore, in the future, a more precise CV setting need to be applied for executing the Tiwary method or other methods.

o The FMO calculations are done on snapshots of the metaMD simulation – most importantly I'd like to see a stable binding mode and a FMO calculation on this. In addition, I do not understand how the FMO calculations can help to solidify the claim of slower off-rates. If the authors would claim stronger binding for different decorations of the phenyl ring, this would be in scope of the theory, but kinetics are clearly not the goal of a FMO calculation

> We apologize our unclear and inappropriate description regarding this point. Indeed, FMO calculation was performed to analyze the stable binding mode (docking structure), as shown in Figure 3e (Fig. 3d in the original manuscript). The primary objective of the FMO calculation was not to determine the kinetics. Our intention is to assess the interaction mode between KSI-6666 and the amino acids within the binding pockets, not to confirm the slow dissociation. To avoid any confusion, we made the necessary modifications to our revised manuscript (Page 14, line 381) and also excluded inappropriate statements regarding this point.

Minor:

- *Caption Figure 3d: The term is “exchange energy” rather than “exclusion energy”*

> Thank you for this notice. We corrected it in the revised manuscript.

Taking all this together, I can clearly not recommend the publication of this manuscript.

Reviewer #3 (Remarks to the Author):

Overall, the article provides detailed insights into the development and characterization of KSI-6666 as a selective antagonist of SIPR1. The compound shows promising potential for the treatment of intestinal inflammation and offers advantages such as persistent efficacy and pseudoirreversible inhibition.

The study is significant to the field and related fields and meets all the expected standards for experimental work. The article highlights the discovery of KSI-6666 as a new pseudoirreversible inhibitor of SIPR1, which exhibited persistent efficacy in vivo. The study also proposes the two-step induced fit model to explain the slow dissociation of KSI-6666 from SIPR1. The role of Met124 in the binding pocket formation and the importance of the benzene ring moiety in KSI-6666 are also noteworthy findings.

A comprehensive methodological approach was used in the study, including molecular dynamics and quantum mechanics methods, along with experimental validation. Methods are well described with enough details for the work to be reproduced.

The data are well analysed and properly presented. Presented data, along with the computational predictions and experimental validations, support the conclusions and claims made regarding the pseudoirreversible inhibition of SIPR1 by KSI-6666 and the role of Met124 in the binding process.

However, there is a lack of comprehensive comparison. The discussion part is poor. The article does not explicitly discuss or compare its findings with the existing literature on SIPR1 inhibition. It would be beneficial to provide a thorough review and comparison of previous studies, highlighting the novelty and significance of the current findings. Also, possible limitations of the study should be identified.

> Thank you for this comment. We added a discussion regarding comparisons with previously developed inhibitors (Page 17, line 443). We also discuss the limitation of this study in Page 18, line 516).

Reviewer #4 (Remarks to the Author):

The manuscript by Maruyama Y et al., aims to find a potent inhibitor of sphingosine 1-phosphate receptor (S1PR1) for treatment of inflammatory disorders. A lead molecule is used to screen candidate inhibitors in vitro and in vivo, to determine the impact on S1PR1 signalling and lymphopenia. The newly identified compound KSI-6666 is then tested in two pre-clinical murine models of inflammatory bowels disease. Administration of the compound significantly reduced inflammation in both models in a similar manner to a known S1PR1 inhibitor, KRP. Molecule modelling of the interaction of KSI-6666 with S1PR1 was then carried out to determine mechanism of inhibition (requirement of a binding of benzene ring KSI-6666 within a methionine in the ligand-binding pocket of S1PR1).

Previous publications have shown that S1PR1 regulated cell migration during inflammation. Several inhibitors for S1PR1 exist, including FTY720, a small molecule that is phosphorylated in vivo to form FTY720-P, binds to S1PR and mediates their internalization and degradation. Although effective in pre-clinical models of renal transplant, this drug elicited a transient, asymptomatic bradycardia in patients (Budde K et al., 2002). This led to the development of novel inhibitors of S1PR1, including molecules with a similar structure to FTY720-P, for example, W146, which had a poor efficacy in vivo, or structurally distinct molecules, for example NIBR-0213. Further refinement of drug design may find more effective molecules for treatment of inflammatory disorders.

I have some concerns over the pre-clinical models and some other major points raised below, which require corrections to the current manuscript before it is acceptable for publication within Nature Communications.

Major comments:

1. For the T-cell transfer model, the purity of transferred T cell populations should be shown in the supplementary Figure. The authors state that they treat with the compounds for 14-16 days two weeks after transfer, a total of 4-weeks for the model. According to published protocols, colitis in this model is readily detectable from 5 weeks, with a weight loss becoming evident 3-5 weeks post-reconstitution (Powrie F et al., 1993). Because of this difference in time-course and the variability of the DAI score given in Figure 2A, it would be useful to see weight loss as an independent Figure.

> Thank you for this advice. We exhibited the weight loss data in Fig. 2c. Regarding the T cell purity, we regret to note the absence of purity assessment for the transferred T cells in this experimental set. The isolation of T cells employed a naive T cell isolation kit obtained from R&D (Cat # MCD45), wherein the manufacturer's protocol specified an expected purity level of CD4⁺/CD62L⁺/CD44^{low} naive T cells as 85-95%. After this separation, CD45RB^{hi} cells were positively selected by the MACS system as described in Methods section. Despite the lack of direct purity verification in this specific experiment, the isolated T cell population was uniformly transferred into recipients. The experimental design aimed to evaluate the efficacy of KSI-6666 in comparison to a control within this set. It is acknowledged that variations in purity may exist between experimental sets; however, it is important to emphasize that the assessment of KSI-6666 efficacy remains robust by comparing results between the control and KSI-6666-treated groups, which received the identical isolated T cell pool. Thus, our approach allows for meaningful conclusions regarding the impact of KSI-6666, notwithstanding minor variations in T cell purity across experimental sets.

In addition, histological score is often comprised of more extensive parameters, including abnormal crypts (score 0–3), presence of crypt abscesses (score 0–1), mucosal erosion and ulceration (score 0–1) and submucosal spread to transmural involvement (score 0–3) (Ostanin D et al., 2009). To support their Figure, the authors should provide several clear images with the histological features highlighted for each treatment group. Authors also should include a statement regarding whether section scoring was blinded and whom this was performed by.

> In the revised manuscript, we used a histological scoring system that incorporates the above-mentioned parameters, as in a previous study (PMID: 30872391). We showed scores with new parameters and typical histological images with higher resolution in Fig. 2c. Also, we described that it was not double blinded but was performed by a veterinary pathologist in Methods section (Page 26, line 736).

Colon weight in Figure 2a should be re-labelled colon length to weight ratio (ratios for animals with colitis are usually between 0.02 and 0.12 g/cm).

> Unfortunately, we did not get colon length data in these experiments. We are sorry for this.

A precedent for using S1009 mRNA/18S rRNA ratio should be established, given this marker is reported to be upregulated during chronic colitis (Brudzewsky D et al., 2009).

> We add a sentence describing the validity using this marker to monitor inflammation (Page 7, line 183).

2. For the oxazolone-induced colitis model, it is more common to see colon length reported as a sign of inflammation and colon shortening, not colon weight (Figure 2C). Weight loss of animals would be useful to see here.

> We showed the weight loss data in Extended Data Fig. 2f. Unfortunately, we did not measure the colon length in this experiment.

Was the colon tissue stimulated before measuring IL-4 protein production? IL-4 production is variable in this model (see Boirivant M et al., 1998).

> We apologize for this confusion caused by our insufficient explanation. This was not an experiment in which T cells were fractionated from LPs and TCR stimulated. Because IL-4 has a critical role in disease progression in this mouse model, the amount of IL-4 protein in the colon tissue was measured by ELISA. This was briefly explained in the text (Page 8, line 184).

Some examples of gross pathology or histology would support conclusions on inhibition of inflammation from the Figures shown.

> Unfortunately, we did not get pathology data in this experiment set. Considering this incompleteness including the lack of some data, we moved these data to Extended Data Fig. 2e.

3. It is not clear why lymphocyte count is shown for cynomolgus monkeys (Figure 1J), with leukocyte count for rats (Figure 1B) and heart rate for guinea pigs (Figure 1G). Some consistency in recordings for both parameters in rodent and primate species is needed. Details of the guinea pigs should be given in the Materials and Methods.

> To detect bradycardia caused by fingolimod-related compounds, it was reported that experiments using guinea pigs reflect the human situation (PMID: 24069292). We briefly explain this point (Page 6, line 145).

To ensure consistency with primate data, an experiment was carried out utilizing rats to assess the impact on blood lymphocyte counts. We exhibited the data in Figure 1i and briefly explained it in the manuscript (Page 6, line 153).

4. Given the variable effect of KSI, particularly the significance in inhibition of inflammation compared to KRP, it is not possible to conclude that the authors have achieved their aim of discovering a novel potent antagonist for the treatment of inflammatory disease.

> Thank you for your comment. We intended to discover a novel 'competitive' S1P1 antagonist for the treatment of inflammatory diseases. We utilized KRP as the control drug because a competitive inhibitor with satisfactory efficacy was not available. Taking your comment into consideration, we have made some modifications to our sentences (Page 2, line 31, Page 4, line 94, Page 5, line 104). Furthermore, while the efficacy of KSI-6666 was more variable than that of KRP in the colitis model, it was comparable to that of FTY720 in the EAE model, which was newly added in the revised manuscript. This also supports our claim of discovering a novel potent competitive antagonist in this study.

5. Clear details of how compound 1 was identified as a lead compound need to be given. This sets the scene for identification of KSI-6666.

> We started first screening from 1181 compounds related to previous S1P1 inhibitors. We mentioned this screening and optimization process (Page 5, line 107).

6. Given the introduction sets a precedent for the identification of new S1PR1 inhibitors for treatment of autoimmune conditions, such as MS, it is not clear why the authors test their compounds in two differing models of inflammatory bowel disease. Neither of these pre-clinical models represent an autoimmune inflammatory condition.

> Thank you for this comment. Rodent EAE data were exhibited in Fig. 2a and b. Comparison between KSI-6666 and fingolimod was performed in Fig. 2b (Page 7, line 171).

7. A clearer definition of a “pseudoirreversible inhibitor” needs to be given. From my understanding this term applies to covalent inactivation of an enzymatic active site, followed by spontaneous hydrolysis, which results in restoration of function. It is not clear from the experiments shown that KSI-6666 fulfils these criteria.

> Thank you for your comment. We understand that pseudoirreversible inhibitory activity refers to a mode of inhibition characterized by an exceptionally slow dissociation of a ligand or inhibitor from its target receptor. In contrast to truly irreversible inhibition, pseudoirreversible inhibitors do not establish a “permanent” covalent bond with the receptor but instead display prolonged and difficult-to-reverse binding interactions. The mechanisms underlying pseudoirreversible inhibition should be diverse. As mentioned by this reviewer, if an inhibitor forms a reversible covalent bond, leading to a slow dissociation rate, such an inhibitor should be classified as a pseudoirreversible inhibitor. In the case of KSI-6666, pseudoirreversible inhibition is achieved through slow dissociation without covalent bond formation. We explain the pseudoirreversible inhibition in Page 4, line 75.

8. It is not clear from the experiments shown, how KSI-6666 can be deemed to have “inverse agonistic activity”. From the definition given, this term applies to spontaneous receptor activation, not competitive inhibition of receptor signaling.

> Generally, an inverse agonist refers to a compound that attaches to the same receptor as an agonist but elicits a pharmacological reaction that is contrary to that of the agonist. For KSI-6666, Fig. 1c in our manuscript show its inverse agonistic activity. We modified sentences for the clarity in Page 5, line 127.

REVIEWER COMMENTS

Reviewer #1 (Remarks to the Author):

The authors have addressed all of my comments with new data and text edits. The work is much improved and the conclusions are better supported by data. In addition, the new in vivo data adds much to the significance of the work.

Reviewer #2 (Remarks to the Author):

I very much appreciate the additional experiments on the receptor half live, the pharmacokinetics and the prolonged in-vitro incubation times. These data solidify the statements on the slow dissociation and the pseudo-irreversibility.

I still, however, don't see any improvement in the modeling procedures, which are not done at (in my opinion) state of the art level. The docked binding pose is now actually shown to be significantly different from the cryo-EM, also in the structurally identical parts (ExtFigure 3), the RMDs of the docked ligand is by far higher than that one of W146 (visible in Fig 3d) and there is a "minimum" during the meta-MD simulations. Therefore, I am still heavily questioning the starting point of the simulations and the analyses which are based on that (like FMO). The meta-MD simulations are not done in a quantitative way, as I requested in my first review and the whole discussion on the reason for the impact of the Met residue is more or less speculative. So, in my opinion, if the authors still decide not to re-do the MDs in a quantitative way, I suggest deleting the following things: The FMO calculations, as they are based on an arbitrary structure, the discussion of the energy profiles of the metaMD and any computational reasoning, why we see such a slow off-rate. What the authors can still keep, is the fact, that their binding mode is OK'ish compared to the cryoEM, the metaMD gives a very nice impression of the geometry of the exit paths and the plain MD simulations show some interactions in the binding site.

With the new data at hand, we see that the residence half-life of KSI-6666 is 9.4h, whereas that for W146 is 0.2h – this is roughly a factor of 50. When comparing the potency of these two ligands, we see a reported potency of about 400-800 nM (depending on the publication and assay) for W146 – that is a factor of 100 in the potency. Therefore, the difference in the off-rates is totally driven by the different potency of the two ligands. There is plenty of literature, that investigates the difference of kinetics that are to be expected (like faster dissociation of weaker binding ligands) versus unexpected kinetics, e.g., caused by shielded hydrogen bonds, where the difference in binding affinity does not translate into the expected difference in the off-rate.

So, to summarize – after I have seen the kinetic data, I have to say that the authors have found an insurmountable antagonist, that is selective and (now) well characterized. The slow off-rate is not

surprising, when compared to the other S1P1R antagonist W146) (this also holds true for mutants, by the way). By the means of MD-simulations they have investigated and identified the exit-channel for both ligands. The rest of the modeling procedure (in its current form) is mostly speculation and should be omitted. I am still in favor of publishing the findings in Nature, but please refrain from making people believe that such rudimentary computational procedures can explain ligand dissociation kinetics quantitatively.

Point-by-point response to Reviewer#2

As we have made significant modifications and omitted some data, all changes are presented in the manuscript text file using track changes in the submitted MS-Word file in addition to yellow highlights in the PDF file. Point-by-point responses are described below.

I very much appreciate the additional experiments on the receptor half live, the pharmacokinetics and the prolonged in-vitro incubation times. These data solidify the statements on the slow dissociation and the pseudo-irreversibility.

I still, however, don't see any improvement in the modeling procedures, which are not done at (in my opinion) state of the art level. The docked binding pose is now actually shown to be significantly different from the cryo-EM, also in the structurally identical parts (ExtFigure 3), the RMDS of the docked ligand is by far higher than that one of W146 (visible in Fig 3d) and there is a "minimum" during the meta-MD simulations. Therefore, I am still heavily questioning the starting point of the simulations and the analyses which are based on that (like FMO).

We acknowledge that the starting point for the simulation using the 3V2Y structure (X-ray) might not be optimal. In contrast, we speculate that utilizing the structure of 7EO4 (Cryo-EM) as a starting point may also not be perfect, as 7EO4 represents an active form. Therefore, we think that both structures would have their own advantages and disadvantages for use in simulations. Nevertheless, because we agree that our MetaMD simulation seems not to be state-of-the-art, we toned down our claims related to MD simulation in the revised manuscript.

In response to each comment, in terms of ligand RMSD (RMSD from their initial docking poses), while KSI-6666 showed higher RMSDs compared to W146 during the early stages of the conventional MD, this result may not be surprising, considering that W146 is the original ligand in 3V2Y. However, towards the end of the conventional MD simulation (450 ns to 500 ns), the average RMSD stabilized at 2.9 Å for KSI-6666 and 2.3 Å for W146. Thus, neither trajectory diverged infinitely, suggesting convergence to stable structures. In addition, we speculate that the presence of a minimum energy state in the metaMD simulation may reflect the existence of two stable binding structures.

The meta-MD simulations are not done in a quantitative way, as I requested in my first review and the whole discussion on the reason for the impact of the Met residue is more or less speculative. So, in my opinion, if the authors still decide not to re-do the MDs in a quantitative way, I suggest deleting the following things: The FMO calculations, as they are based on an arbitrary structure, the discussion

of the energy profiles of the metaMD and any computational reasoning, why we see such a slow off-rate. What the authors can still keep, is the fact, that their binding mode is OK'ish compared to the cryoEM, the metaMD gives a very nice impression of the geometry of the exit paths and the plain MD simulations show some interactions in the binding site.

We acknowledge that our metaMD calculation is fundamental as compared to Tiwary's methods suggested by this reviewer. Therefore, in light of the reviewer's feedback, we have opted to exclude FMO calculations, because it was built upon the output from the metaMD calculations. Moreover, the findings from the metaMD calculations have been largely tone-downed, recognizing their inconclusive nature. Thus, we describe that the metaMD data provided some information regarding the dissociation process of KSI-6666, but not conclusive findings related to the slow dissociation kinetics, in the revised manuscript.

However, we would like to emphasize that this fundamental metaMD calculation has yielded a speculation that should be validated by some experiments. Indeed, the significance of Met124 in the pseudo irreversible inhibition of KSI-6666 would have eluded us without the metaMD calculation. Therefore, speculations from metaMD calculations that align with experimental observations have been revised and retained.

With the new data at hand, we see that the residence half-life of KSI-6666 is 9.4h, whereas that for W146 is 0.2h – this is roughly a factor of 50. When comparing the potency of these two ligands, we see a reported potency of about 400-800 nM (depending on the publication and assay) for W146 – that is a factor of 100 in the potency. Therefore, the difference in the off-rates is totally driven by the different potency of the two ligands. There is plenty of literature, that investigates the difference of kinetics that are to be expected (like faster dissociation of weaker binding ligands) versus unexpected kinetics, e.g., caused by shielded hydrogen bonds, where the difference in binding affinity does not translate into the expected difference in the off-rate.

Given the difference in the potency between KSI-6666 and W146, this variance may influence the off-rate experiment. Yet, upon juxtaposing compound 4 with KSI-6666 (Figure 7), where the disparity in potency approximates a factor of 2, the difference in dissociation half-life emerges notably larger, spanning approximately a factor of 10 (0.96 versus 9.4 hours). Thus, we think that the slow dissociation kinetics of KSI-6666 elude sole explanation through strong potency alone. However, considering the fundamental nature of our computer calculation, detailed mechanisms remain unclear. Therefore, we have included the limitations of our study in Discussion (Page 16, line 28)

So, to summarize – after I have seen the kinetic data, I have to say that the authors have found an insurmountable antagonist, that is selective and (now) well characterized. The slow off-rate is not surprising, when compared to the other SIP1R antagonist W146) (this also holds true for mutants, by the way). By the means of MD-simulations they have investigated and identified the exit-channel for both ligands. The rest of the modeling procedure (in its current form) is mostly speculation and should be omitted. I am still in favor of publishing the findings in Nature, but please refrain from making people believe that such rudimentary computational procedures can explain ligand dissociation kinetics quantitatively.

Thank you for acknowledging and agreeing with our findings. Again, we acknowledge that our metaMD calculation may be rudimentary and straightforward. Nevertheless, we wish to emphasize that despite its fundamental nature, this metaMD calculation has generated speculations that warrant validation through experiments.

In summary, in response to the reviewer's feedback, we have decided to omit data regarding FMO calculations, as they rely on output from the metaMD calculations. Furthermore, we have moderated the interpretation of the metaMD findings, acknowledging their inconclusive nature. Additionally, the PLIF analysis data in MOE have been relocated to Extended Data Fig. 5, and the associated discussion has been removed. Moreover, we added a brief discussion regarding a limitation of our simulation study (Page 16, line 28). Conversely, predictions validated by experimental observations arising from the metaMD calculations have been revised and retained. With these improvements, we expect that people will recognize that these simple MD calculations are not sufficient to predict kinetics, and in this case, that experimental verification of the speculation obtained from the calculations was performed.

REVIEWERS' COMMENTS

Reviewer #2 (Remarks to the Author):

The current version of the manuscript is OK in terms of the use and interpretation of computational results. It can be published in its current form from my point of view.